# During heat stress in *Myxococcus xanthus*, the CdbS PilZ domain protein, in concert with two PilZ-DnaK chaperones, perturbs chromosome organization and accelerates cell death

**Michael Seidel**[1], **Dorota Skotnicka**[1], **Timo Glatter**[2], **Lotte Søgaard-Andersen**[1]*

**1** Department of Ecophysiology, Max Planck Institute for Terrestrial Microbiology, Marburg, Germany, **2** Core Facility for Mass Spectrometry & Proteomics, Max Planck Institute for Terrestrial Microbiology, Marburg, Germany

* sogaard@mpi-marburg.mpg.de

## Abstract

C-di-GMP is a bacterial second messenger that regulates diverse processes in response to environmental or cellular cues. The nucleoid-associated protein (NAP) CdbA in *Myxococcus xanthus* binds c-di-GMP and DNA in a mutually exclusive manner *in vitro*. CdbA is essential for viability, and CdbA depletion causes defects in chromosome organization, leading to a block in cell division and, ultimately, cell death. Most NAPs are not essential; therefore, to explore the paradoxical *cdbA* essentiality, we isolated suppressor mutations that restored cell viability without CdbA. Most mutations mapped to *cdbS*, which encodes a stand-alone c-di-GMP binding PilZ domain protein, and caused loss-of-function of *cdbS*. Cells lacking CdbA and CdbS or only CdbS were fully viable and had no defects in chromosome organization. CdbA depletion caused post-transcriptional upregulation of CdbS accumulation, and this CdbS over-accumulation was sufficient to disrupt chromosome organization and cause cell death. CdbA depletion also caused increased accumulation of CsdK1 and CsdK2, two unusual PilZ-DnaK chaperones. During CdbA depletion, CsdK1 and CsdK2, in turn, enabled the increased accumulation and toxicity of CdbS, likely by stabilizing CdbS. Moreover, we demonstrate that heat stress, possibly involving an increased cellular c-di-GMP concentration, induced the CdbA/CsdK1/CsdK2/CdbS system, causing a CsdK1- and CsdK2-dependent increase in CdbS accumulation. Thereby this system accelerates heat stress-induced chromosome mis-organization and cell death. Collectively, this work describes a unique system that contributes to regulated cell death in *M. xanthus* and suggests a link between c-di-GMP signaling and regulated cell death in bacteria.

## Author summary

The nucleotide-based second messenger c-di-GMP in bacteria controls numerous processes in response to environmental or cellular cues. Typically, these processes are related to lifestyle transitions between motile and sessile behaviors. However, c-di-GMP also

**Data Availability Statement:** The authors declare that all data supporting this study are available

within the article and its Supplementary Information files.

**Funding:** This work was supported by the Max Planck Society (to LS-A). The funders had no role in study design, data collection and analysis, decision to publish, or preparation of the manuscript.

**Competing interests:** The authors have declared that no competing interests exist.

regulates other processes. In *Myxococcus xanthus*, CdbA is a DNA-binding and nucleoid-associated protein that helps to organize the large chromosome. CdbA binds c-di-GMP and DNA in a mutually exclusive manner. While other nucleoid-associated proteins are not essential, CdbA is essential. Here, we show that the crucial function of CdbA is to maintain the level of the c-di-GMP-binding PilZ-domain protein CdbS appropriately low. The CdbS level is not only increased upon depletion of CdbA but also in response to heat stress. Under both conditions, the increased CdbS level perturbs chromosome organization and ultimately causes cell death. The CdbA/CdbS system represents a unique system that contributes to regulated cell death in *M. xanthus* and suggests a link between c-di-GMP signaling and regulated cell death.

## Introduction

Bis-(3'-5')-cyclic dimeric GMP (c-di-GMP) is a versatile, ubiquitous nucleotide-based second messenger in bacteria involved in regulating diverse processes in response to environmental or cellular cues [1,2]. The balance between the opposing activities of c-di-GMP synthesizing diguanylate cyclases and c-di-GMP-degrading phosphodiesterases determines the cellular c-di-GMP level [1,2]. In addition to their enzymatically active domain, these enzymes often contain sensory domains that enable their regulation in response to specific cues [1,2]. c-di-GMP binds to and allosterically regulates effectors to implement downstream responses at the transcriptional, translational or post-translational level [1,2]. While DGCs and PDEs contain conserved domains, c-di-GMP-binding effector proteins are highly diverse and include, e.g. PilZ-domain proteins [3–7] and members of diverse transcription factor families as well as a nucleoid-associated protein [8–14]. C-di-GMP regulates many diverse processes that are typically associated with lifestyle transitions, including biofilm formation, motility, adhesion, synthesis of secreted polysaccharides, cell cycle progression, development and virulence [1,2]. Thereby, c-di-GMP contributes to fitness but is not essential. We recently reported an unexpected link between c-di-GMP and chromosome organization in *Myxococcus xanthus* mediated by CdbA, a c-di-GMP binding and nucleoid-associated protein (NAP) [11]. Typically, NAPs are small, abundant proteins that bind DNA with moderate sequence specificity causing bending, wrapping or bridging of DNA, thereby contributing to chromosome topology and organization [15–17]. Generally, NAPs have minor effects on transcription and are not essential [15–17]; however, CdbA is essential [11]. Here, we focused on understanding the mechanism underlying CdbA essentiality.

*M. xanthus* is a Gram-negative bacterium with a nutrient-regulated biphasic lifecycle [18,19]. Under nutrient-replete conditions, the rod-shaped cells proliferate and, using gliding motility and type IV pili (T4P)-dependent motility, generate coordinately spreading colonies in which cells also prey on other microbes. In response to nutrient depletion, a developmental program initiates that culminates in the formation of multicellular, spore-filled fruiting bodies. C-di-GMP accumulates during growth and at a ~10-fold higher level during development [20,21]. During growth, c-di-GMP regulates T4P-dependent motility and the composition of the extracellular matrix including synthesis of the secreted polysaccharide referred to as exopolysaccharide [20,22,23]. During development, the increased c-di-GMP level is essential for fruiting body formation and sporulation [21,24]. Several c-di-GMP binding effectors involved in implementing these responses have been characterized [21–23].

The rod-shaped *M. xanthus* cells divide at midcell between two fully segregated chromosomes [25,26]. Each daughter cell contains a single, fully replicated chromosome with the *ori*

and *ter* regions anchored in the subpolar regions close to the old and new cell pole, respectively by a scaffold composed of the BacNOP bactofilins and the PadC adaptor [27–30]. Chromosome replication is initiated once per cell cycle and soon after cell division [29]. Segregation occurs in parallel with replication and depends on a classical ParAB*S* system, in which ParB binds to centromere-like *parS* sites close to the *ori* while the ParA ATPase mediates segregation [29–31]. Segregation also depends on the structural maintenance of chromosome (SMC) complex, which contributes to separating the two daughter chromosomes [32,33]. During segregation, one ParB/*parS* nucleoprotein complex remains in the subpolar region of the old pole while the second copy translocates to the subpolar region of the opposite pole [29].

CdbA and its paralog CdbB belong to the widespread ribbon-helix-helix superfamily of DNA binding protein, and all fully sequenced Myxococcales genomes encode at least one ortholog [11,34]. While CdbA is essential for viability, CdbB is not [11]. CdbA depletion results in disrupted chromosome organization and impeded chromosome segregation, causing cell division defects, cellular filamentation, and eventually, cell lysis and death [11]. CdbA and CdbB bind c-di-GMP *in vitro* [11]. The binding of c-di-GMP does not alter the tetrameric state of CdbA but its conformation [11]. Importantly, DNA and c-di-GMP binding by CdbA involve the same interface of the tetramer and is mutually exclusive *in vitro* [11]. Consistently, CdbA variants that cannot bind c-di-GMP do not bind DNA *in vitro* and, consequently, these variants are non-functional *in vivo* and do not support proper chromosome organization [11]. In ChIP-seq analyses, CdbA binds >550 sites in the *M. xanthus* genome with moderate sequence specificity [11]. However, CdbA depletion causes no or only minor changes in transcription [11]. Based on these observations, and because CdbA is highly abundant with an average cell containing ~7000 CdbA monomers, we suggested that CdbA is an essential, ligand-regulated NAP and that c-di-GMP modulates DNA binding by CdbA [11]. According to this model, the primary function of CdbA is in chromosome organization, thereby supporting chromosome segregation, and the inhibition of cell division is a secondary effect caused by the defects in chromosome organization and segregation [11].

To investigate the mechanism underlying CdbA essentiality, we isolated suppressor mutations that restored cell viability without CdbA. Most mutations mapped to a gene that encodes a c-di-GMP-binding PilZ domain protein, which we named CdbS, and caused loss-of-function of *cdbS*. Cells lacking CdbA and CdbS or only CdbS were fully viable and had no evident defects in chromosome organization. CdbA depletion increased CdbS accumulation dependent on two unusual PilZ-DnaK chaperones, which we named CsdK1 and CsdK2. Furthermore, an increased CdbS accumulation was sufficient to disrupt chromosome organization and, ultimately, cause cell death. We identify heat stress as a physiological cue causing increased CdbS accumulation in a CsdK1 and CsdK2-dependent manner, thereby contributing to chromosome mis-organization and cell death in response to heat stress.

## Results

### Isolation of suppressor mutants that are viable in the absence of CdbA

To investigate how the lack of CdbA is toxic to cells, we sought spontaneous suppressor mutants that were viable without CdbA. To this end, we used a strain with an in-frame deletion of native *cdbA* (Δ*cdbA*) and in which ectopic expression of an active *cdbA-mCherry* (from here on CdbA-mCh) fusion is controlled by a vanillate-inducible promoter ($P_{van}$) [11]. Even when $P_{van}$ is maximally induced in this strain in the presence of 500μM vanillate, the cellular level of CdbA-mCh is lower than when *cdbA-mCherry* is expressed from the native site and, therefore, this strain has a lower growth rate than the *cdbA*[+] WT strain [11]. Upon removal of vanillate, and correlating with the earliest time-point at ~24hrs at which CdbA-mCh is no

**Table 1. Mutations identified by whole genome sequencing of suppressors of CdbA-mCh essentiality.**

| Suppressor mutant | Gene locus | Mutation | Position of mutation | Amino acid change[1] |
|---|---|---|---|---|
| #2 | *mxan_4328* (*cdbS*) | C → T | 60bp upstream of transcriptional start site | NA |
| | *mxan_3823* | G → T | 261bp downstream of first nucleotide in start codon | T → T (silent) |
| #5 | *mxan_4328* (*cdbS*) | A → C | 95bp downstream of first nucleotide in start codon | V32G |
| #8 | *mxan_4328* (*cdbS*) | T → C | 67bp upstream of transcriptional start site | NA |
| #12 | *mxan_4362* (*cdbB*) | A → G | 65 bp downstream of first nucleotide in start codon | Q22R |
| #14 | *mxan_4328* (*cdbS*) | G → A | 226bp downstream of first nucleotide in start codon | Q76stop |
| #15 | *mxan_4328* (*cdbS*) | A → C | 95bp downstream of first nucleotide in start codon | V32G |
| #18 | *mxan_4328* (*cdbS*) | T → C | 67bp upstream of transcriptional start site | NA |
| #19 | *mxan_4328* (*cdbS*) | A → C | 95bp downstream of first nucleotide in start codon | V32G |
| | *mxan_3618* | A → C | 640bp downstream of first nucleotide in start codon | A → A (silent) |

[1] NA, not applicable

longer detectable by immunoblotting, growth of this strain arrests and cells eventually lyse [11]. After plating cells on CTT broth without vanillate, 20 independent mutants were isolated that grew in the absence of vanillate. Sequence analysis demonstrated that six mutants had mutations in the vanillate-inducible promoter or in the gene encoding the vanillate repressor. By immunoblot analysis, six of the remaining 14 strains accumulated CdbA-mCh without vanillate. The remaining eight strains did not accumulate CdbA-mCh in the absence of vanillate and, thus, were viable without CdbA-mCh. Whole genome sequencing revealed that one suppressor strain had a mutation in *cdbB*, which encodes the paralog of CdbA, resulting in a Q22R substitution in CdbB (Table 1). The seven remaining suppressor strains had four different mutations in *mxan_4328* (Fig 1A and Table 1). *Mxan_4328* encodes a stand-alone PilZ domain protein (Fig 1A and 1B) [23] and is not part of an operon [24] (S1A Fig). From here on, we focus on *mxan_4328*, which we named *cdbS* (CdbA essentiality suppressor).

## The PilZ-domain protein CdbS binds c-di-GMP

The CdbS PilZ domain contains the conserved bipartite c-di-GMP binding motif characteristic of PilZ domains [2] (Fig 1B). A high-confidence AlphaFold-based structural model predicts monomeric CdbS as a six-stranded β-barrel typical of PilZ domains, but the α-helix at the C-terminus characteristic of canonical PilZ domains was not predicted (Figs 1C and S2A), classifying CdbS as an xPilZ domain protein [35]. CdbS is conserved in the *Cystobacterineae* and *Nannocystineae* suborders but not in the *Sorangineae* suborder of the Myxococcales [23] (S1B and S1C Fig). The genetic neighbourhood of *cdbS* is conserved but encodes neither proteins with a predicted function in chromosome organization and segregation nor in cell division (S1C Fig). A systematic analysis of PilZ domain proteins in *M. xanthus* previously demonstrated that CdbS is dispensable for viability, motility, exopolysaccharide synthesis, and development [23].

We tested whether CdbS binds c-di-GMP *in vitro* using purified His$_6$-CdbS and His$_6$-CdbS$^{R9A}$, which contains the Arg9Ala substitution in the c-di-GMP binding motif and abolishes c-di-GMP binding by the PilZ domain protein YcgR [6] (Figs 1B and S2B). Using bio-layer interferometry with a streptavidin sensor on which biotinylated c-di-GMP was immobilized, titration experiments revealed that His$_6$-CdbS binds c-di-GMP with a K$_d$ of ~1.4μM, while His$_6$-CdbS$^{R9A}$ did not detectably bind c-di-GMP (Fig 1D and 1E). This K$_d$ is similar to that of other c-di-GMP binding PilZ domains [3,5,6]. The estimated cellular concentration of c-di-GMP in *M. xanthus* is ~1.4μM [11,20], suggesting that CdbS binds c-di-GMP *in vivo*.

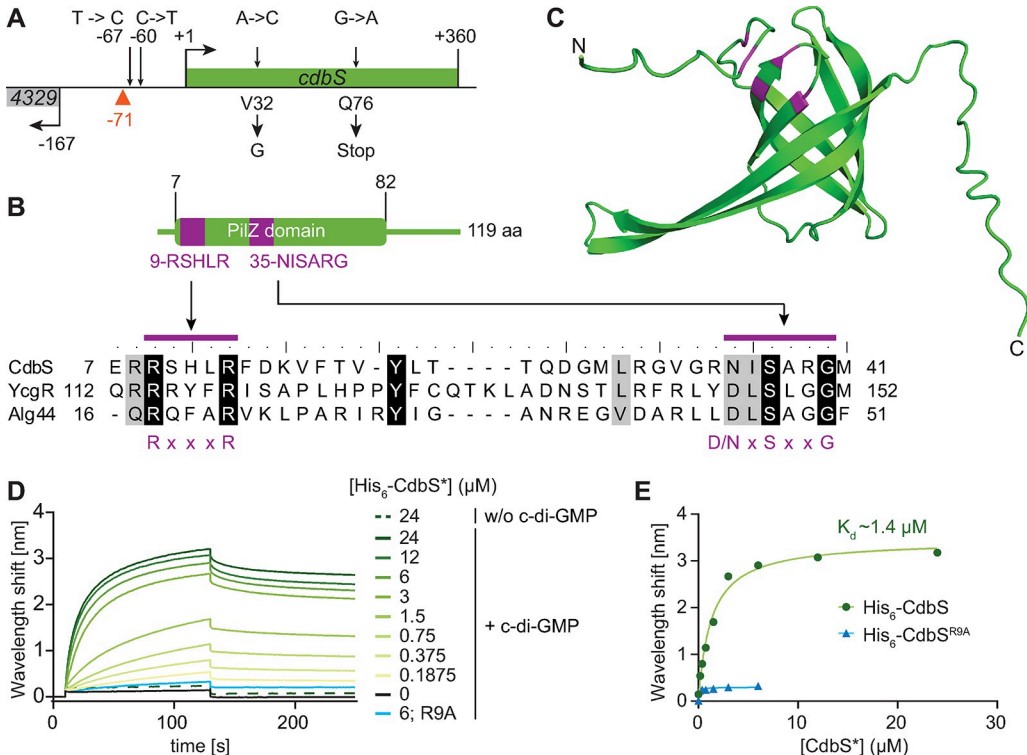

**Fig 1. CdbS is a stand-alone PilZ domain protein and binds c-di-GMP. A.** *cdbS* locus. Kinked arrows indicate transcription start sites and +1 the transcription start site of *cdbS* [24]. Suppressor mutations upstream of and within *cdbS* are indicated by arrows above and amino acid substitutions/stop codon below. The red triangle indicates the CdbA peak summit at -71 from a ChIP-seq analysis in which an active CdbA-FLAG protein was used as bait [11]. **B.** CdbS is a stand-alone PilZ domain protein. The PilZ domain encompasses residues 7–82. The residues marked in purple indicate the bipartite c-di-GMP binding motif. Below, alignment of residues 7–41 of CdbS with the corresponding residues of the PilZ domains of *E. coli* YcgR [6] and *P. aeruginosa* Alg44 [5]. Residues marked with purple bars indicate the bipartite c-di-GMP binding motif with the consensus below [2]. **C.** AlphaFold model of CdbS. CdbS was modeled as a monomer. Model rank 1 is shown with the conserved residues in the bipartite c-di-GMP binding motif marked in purple. **D.** Bio-layer interferometric analysis of the interaction between CdbS variants and c-di-GMP. Streptavidin coated sensors were loaded with biotinylated c-di-GMP and probed with the indicated concentrations of $His_6$-CdbS (shades of green) or $His_6$-CdbS$^{R9A}$ (blue). The interaction kinetics were followed by monitoring the wavelength shifts resulting fr changes in the optical thickness of the sensor surface during association or dissociation of the analyte. Unspecific binding of $His_6$-CdbS to the sensor was tested in the absence of c-di-GMP (w/o c-di-GMP) and is shown as the dashed line. **E.** Analysis of the binding data shown in panel D. Plot shows the equilibrium levels measured at the indicated $His_6$-CdbS* concentrations. The data were fitted to a non-cooperative one-site specific-binding model. The calculated $K_d$ for $His_6$-CdbS is shown in the graph.

## Essentiality of CdbA depends on CdbS

Two of the *cdbS* suppressor mutations map upstream of the *cdbS* transcription start site, one results in a premature stop codon at Gln76, and one in a Val32Gly substitution (Figs 1A, S1C and Table 1). We, therefore, hypothesized that the suppressor mutations cause loss-of-function of *cdbS*. To test this idea, we generated an in-frame deletion in *cdbS* (Δ*cdbS*) in the Δ*cdbA* strain in which *cdbA-mCh* is expressed from P$_{van}$. To follow chromosome organization, all strains ectopically expressed a *parB-YFP* fusion from the native promoter integrated in a single copy at the Mx8 *attB* site, generating merodiploid *parB*$^+$/*attB*::P$_{nat}$*parB-YFP* strains.

As reported [11], the CdbA-mCh depletion strain grew when supplemented with vanillate and at a lower rate than WT, and removal of vanillate caused (1) growth arrest after ~24hrs followed by a decrease in the optical density at 550nm (OD$_{550}$) indicating cell lysis, and (2) a three-log defect in plating efficiency (Fig 2A). Importantly, cells of the CdbA-mCh depletion

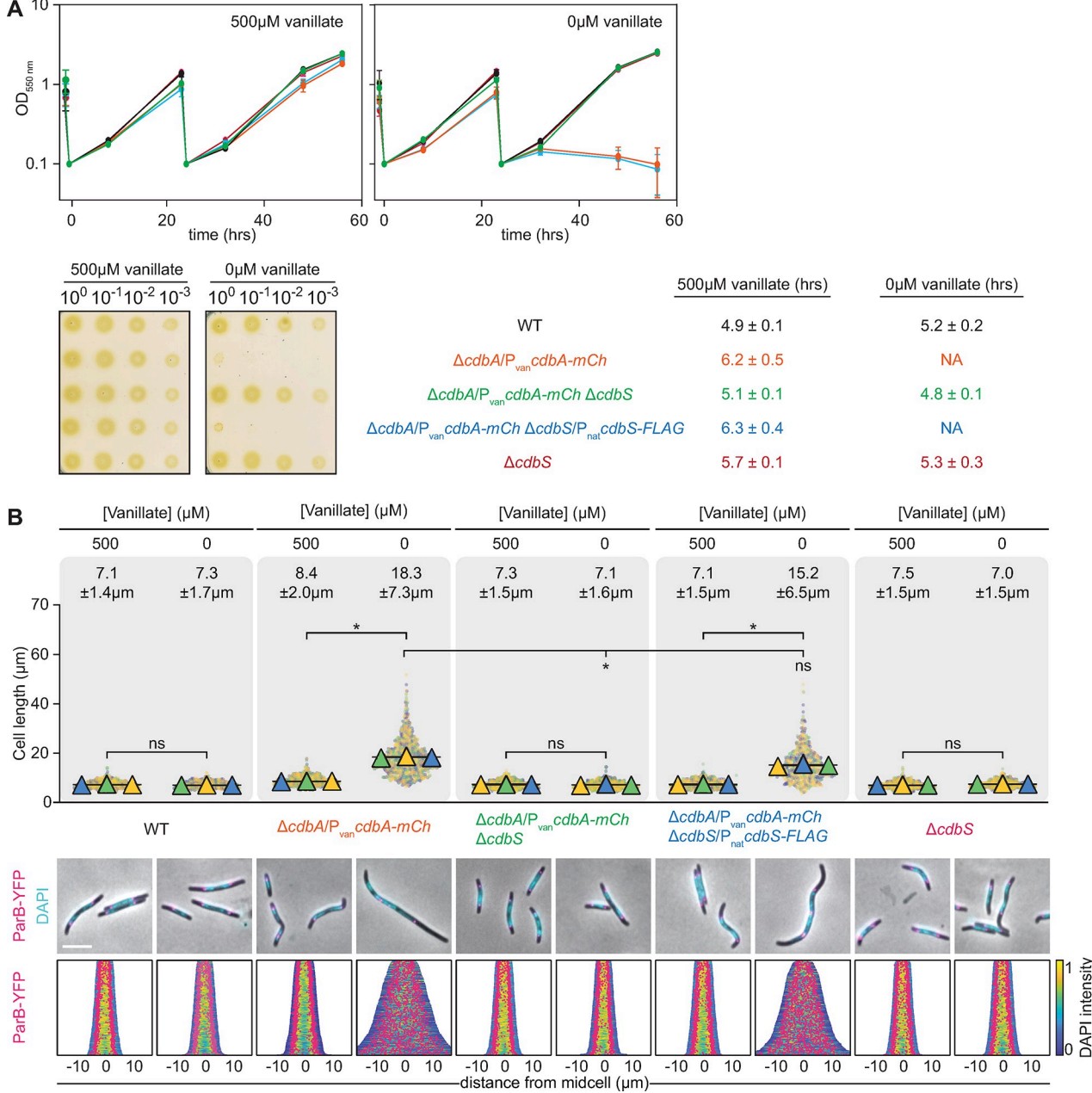

**Fig 2. Lack of CdbS suppresses CdbA essentiality. A.** Growth of strains of indicated genotypes. Cells were grown in 1% CTT broth in suspension culture (upper) or on 1% CTT broth, 1.5% agar on a solid surface (lower) in the presence and absence of vanillate as indicated. Plates were incubated for 96hrs before imaging. Color code used for the growth curves are as in the table. Table indicates generation times as mean ± standard deviation (STDEV) from three biological replicates; NA, not applicable. **B.** Cell length and chromosome organization of strains of indicated genotypes. Cells were grown in the presence and absence of vanillate as indicated. In the absence of vanillate, cells were analyzed 24hrs after removal of vanillate. Cell length measurements are included from three independent experiments indicated in different colored triangles and the mean is based on all three experiments. Numbers above indicate cell length as mean ± STDEV from all three experiments. * $P < 0.0001$, ns, not significant in 2way ANOVA multiple comparisons test. Total number of cells analyzed: 440–870. Lower diagrams, fluorescence microscopy images of cells stained with DAPI and synthesizing ParB-YFP. In the demographs, cells are sorted according to length, DAPI signals are shown according to the intensity scale, and ParB-YFP signals in pink. Scale bar, 5μm. N = 400 cells for all strains. In A and B, all strains are $parB^{+}/parB\text{-}YFP$ merodiploid.

strain additionally containing the Δ*cdbS* mutation were viable in the absence of vanillate and had a growth rate similar to WT (Fig 2A). Moreover, when a *cdbS-FLAG* allele was expressed ectopically under the control of the native promoter ($P_{nat}$) from a single copy integrated at the Mx8 *attB* site in the Δ*cdbS* CdbA-mCh depletion strain, this strain phenocopied the *cdbS*+ CdbA-mCh depletion strain and had a growth defect in the absence of vanillate (Fig 2A). Thus, CdbS-FLAG is active and complements the Δ*cdbS* mutation. Finally, the Δ*cdbS* mutation in the presence of CdbA did not affect viability and growth in agreement with previous findings [23] (Fig 2A). Thus, the toxicity caused by lack of CdbA is suppressed by the lack of CdbS. In other words, the essentiality of CdbA depends on the presence of CdbS, while the lack of CdbS by itself affects neither growth nor viability.

As reported [11], the CdbA-mCh depletion strain supplemented with vanillate had a slightly increased cell length compared to WT, while cells depleted of CdbA-mCh for 24hrs were filamentous (Fig 2B). Importantly, Δ*cdbS* cells depleted of CdbA-mCh had a cell length similar to WT. Moreover, cells were filamentous without vanillate upon ectopic expression of active *cdbS-FLAG* in this strain. Finally, the Δ*cdbS* mutation in the presence of CdbA did not affect cell length.

To assess chromosome organization, cells were stained with 4′,6-diamidino-2-phenylindole (DAPI). As reported [25], short WT cells had one nucleoid centered around midcell, while longer cells had two well-separated nucleoids centered around the quarter cell length positions (Fig 2B). Furthermore, the ParB-YFP cluster(s) localized in the subpolar region(s) and close to the outer edges of the nucleoid(s) (Fig 2B). The same pattern was observed for the CdbA-mCh depletion strain in the presence of vanillate. However, and as reported [11], after 24hrs in the absence of vanillate, and thus depleted of CdbA-mCh, cells had an irregular distribution of the nucleoids along the cell length and the nucleoids appeared more decondensed than in WT (Fig 2B). Moreover, the ParB-YFP clusters were irregularly distributed over the nucleoid masses along the cell length and no longer restricted to the subpolar region(s) (Fig 2B). Strikingly, Δ*cdbS* cells depleted of CdbA-mCh had nucleoids and ParB-YFP clusters organized as in WT (Fig 2B). Notably, ectopic expression of CdbS-FLAG in these cells resulted in the irregular distribution of decondensed nucleoids as well as ParB-YFP clusters along the cell length (Fig 2B). Finally, cells lacking only CdbS, had nucleoids and ParB-YFP clusters organized as in WT, as expected from the lack of a growth defect (Fig 2B).

We conclude that lack of CdbA *per se* does not result in the disruption of chromosome organization, i.e. the irregular distribution of decondensed nucleoids and ParB-YFP clusters along the cell length, with subsequent cellular filamentation and cell death. Instead, these defects are only observed when CdbA is depleted in the presence of CdbS. Thus, CdbS is the critical factor during CdbA depletion that mediates the disrupted chromosome organization, resulting in cellular filamentation and cell death. Equally, the observation that lack of CdbS in otherwise WT affects neither the chromosome nor viability demonstrates that CdbS is not essential for these two processes. These observations support a model in which a lack of CdbA unleashes CdbS activity.

## Overexpression of *cdbS* phenocopies CdbA-depletion

To understand the connection between CdbA and CdbS, we asked whether CdbA regulates *cdbS* expression. In published data [11], CdbA has a ChIP-seq peak centered at position -71 relative to the *cdbS* transcription start site at +1 (Fig 1A). Nevertheless, in RT-qPCR analyses, the *cdbS* transcript level in the CdbA-mCh depletion strain in the presence and absence of CdbA-mCh was not significantly different (Fig 3A).

To determine whether CdbA regulates CdbS accumulation post-transcriptionally, we generated strains synthesizing CdbS-FLAG from the native site. In the CdbA-mCh depletion

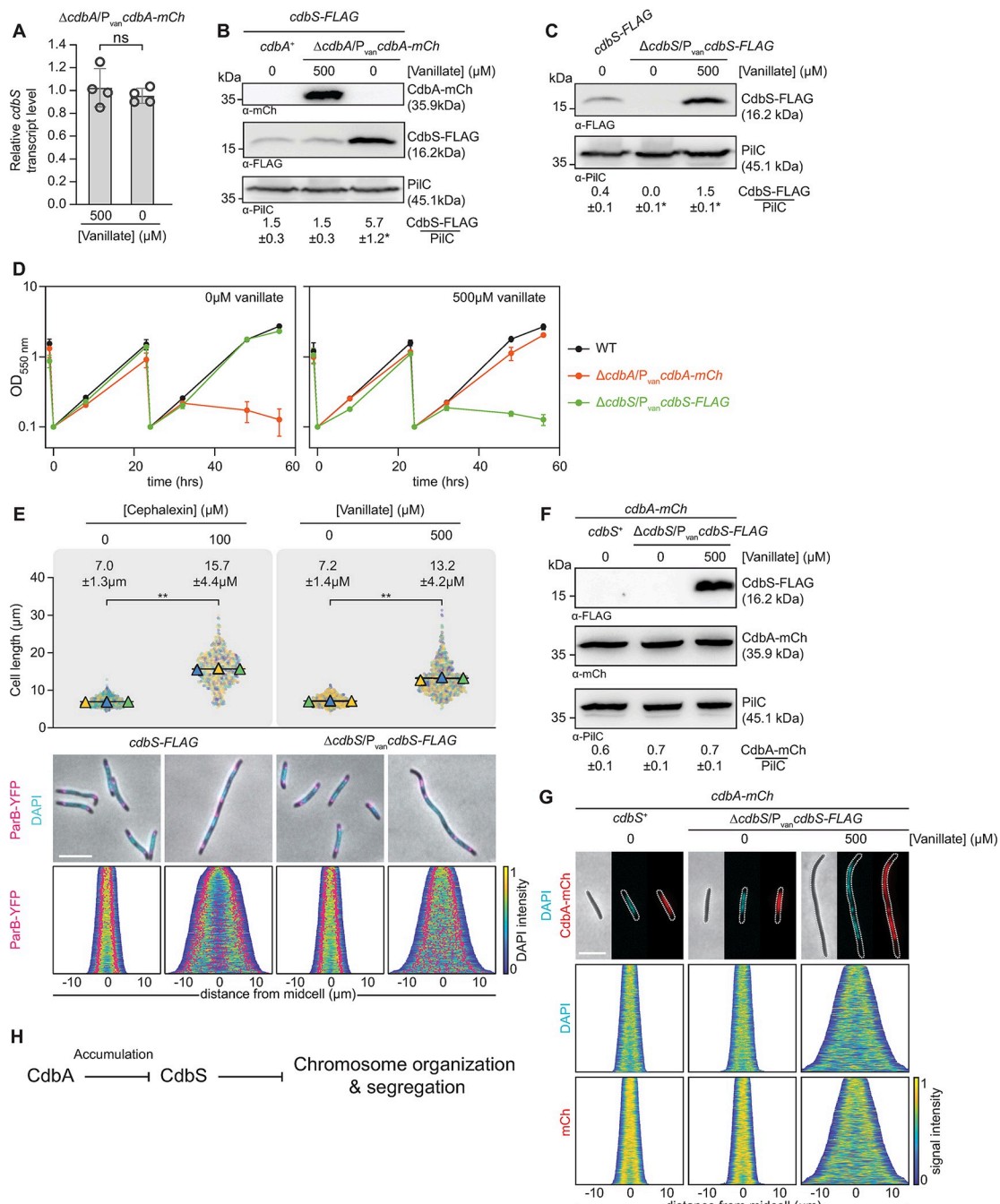

**Fig 3. Overexpression of *cdbS* phenocopies CdbA-depletion. A.** RT-qPCR analysis of *cdbS* expression in the presence and absence of CdbA-mCh. Cells of the indicated genotypes were grown in the presence of vanillate or in its the absence for 24hrs. Transcript levels are shown relative to the level in the presence of vanillate and as mean ± STDEV from four biological replicates with three technical replicates each. Individual data points are shown in black. ns, no significant difference in two-sided Student's t-test. **B.** Immunoblot analysis of CdbS-FLAG accumulation in the presence and absence of CdbA-mCh. Cells of the indicated genotypes were grown in the presence or absence of vanillate as in A. The same amount of total protein was loaded per lane. PilC is used as a loading control. Numbers below show the mean level of CdbS-FLAG normalized by the PilC level ± STDEV calculated from three independent experiments. *, *P*<0.05 in Student's t test in which samples were compared to the *cdbA*⁺ strain. **C.** Immunoblot analysis of CdbS-FLAG accumulation. Cells of the indicated genotypes were grown in the absence of vanillate or in its presence for 24hrs. The *cdbS-FLAG* strain expresses this allele from the native site. Samples were loaded and analyzed as in B. *, *P*<0.05 in Student's t test in which samples were compared to CdbS-FLAG expressed from the native site. **D.** Growth of strains of indicated genotypes. Cells were grown in 1% CTT broth in suspension culture in the presence and absence of vanillate as indicated. The growth curves were prepared from three biological replicates, error bars, mean ± STDEV. **E.** Cell length and

chromosome organization of strains of indicated genotypes. Cells were grown as in C. Cells grown in the presence of cephalexin were analyzed after 8hrs. Cells were analyzed as in Fig 2B. Numbers above indicate cell length as mean ± STDEV from all three experiments. **, $P<0.0001$, ns, not significant in 2way ANOVA multiple comparisons test. Total number of cells analyzed: 497–794. Lower panels, scale bar, 5μm. N = 400 cells for all strains. **F.** Immunoblot analysis of CdbA-mCh accumulation in strains with varying CdbS levels. Cells were grown as in C and samples loaded and analyzed as in B based on three biological replicates. **G.** Phase contrast and fluorescence microscopy images of cells of the indicated genotypes stained with DAPI and expressing CdbA-mCh. Cells were grown as in C. In the demographs, cells are sorted by cell length, DAPI and mCh signals are shown according to the intensity scale. Scale bar, 5μm. N = 400 cells for all strains. **H.** Genetic pathway for the CdbA CdbS interaction. See text for details. In A-G, all strains are *parB⁺/parB-YFP* merodiploid.

strain in the presence of vanillate, CdbS-FLAG accumulated at a level similar to that of a *cdbA⁺* strain in immunoblots with α-FLAG antibodies. But upon depletion of CdbA-mCh, the CdbS-FLAG level increased ~4-fold (Fig 3B). Because *cdbS* transcription is not increased upon CdbA-mCh depletion, we conclude that the increased CdbS level is the result of either increased *cdbS* translation or increased CdbS stability.

To examine whether an increased CdbS level is sufficient to disrupt chromosome organization and cause cell death, we ectopically expressed *cdbS-FLAG* under the control of $P_{van}$ in a $\Delta cdbS$ strain. In the absence of vanillate, CdbS-FLAG was not detectable in immunoblots with α-FLAG antibodies (Fig 3C), cells had a growth rate and a mean cell length similar to WT, and the nucleoid as well as ParB-YFP clusters were organized as in WT (Fig 3D and 3E). In the presence of vanillate for 24hrs, CdbS-FLAG accumulation was ~4-fold increased relative to the strain expressing *cdbS-FLAG* from the native site (Fig 3C). Notably, at ~24hrs growth was arrested followed by a decrease in $OD_{550}$ indicating cell death, cells were filamentous, and the nucleoid as well as ParB-YFP cluster localization were highly disorganized with an irregular distribution of decondensed nucleoids along the cell length and ParB-YFP clusters irregularly distributed over the nucleoid masses and no longer restricted to the subpolar region(s) (Fig 3D and 3E). Increasing or decreasing the cellular CdbS-FLAG level changed neither the cellular level of CdbA-mCh nor its localization over the nucleoid (Fig 3F and 3G). Moreover, ~4-fold overexpression of the non-c-di-GMP binding CdbS^R9A-FLAG variant phenocopied cells overexpressing CdbS-FLAG (S3A and S3B Fig). Thus, cells overexpressing CdbS-FLAG ~4-fold phenocopy cells depleted of CdbA-mCh and, under these conditions, the effect of CdbS-FLAG overexpression is independent of c-di-GMP binding.

In control experiments, we treated WT with cephalexin to specifically inhibit cell division but not chromosome organization [25]. After for 8hrs, cells had elongated and had two to four well-separated nucleoids arranged regularly along the cell length, as well as an increased number of ParB-YFP clusters, arranged evenly along the outer edges of the nucleoids (Fig 3E). These observations support that the primary defect in cells overexpressing CdbS-FLAG, similar to cells depleted of CdbA-mCh, is in chromosome organization, not cell division.

Altogether, we conclude that an elevated CdbS level is sufficient to disrupt chromosome organization with the irregular distribution of decondensed nucleoids and ParB-YFP clusters along the cell length, thereby causing cellular filamentation and cell death. Because this phenotype mirrors the phenotype caused by CdbA depletion, these observations suggest a genetic pathway in which CdbA inhibits the accumulation of CdbS (Fig 3H). CdbA depletion alleviates this inhibition, and CdbS accumulates at an increased level. The increased CdbS level, in turn, interferes with chromosome organization, thereby inhibiting cell division and causing cellular filamentation and, eventually, cell death.

## CdbS pulls down the DnaB helicase, chaperones and co-chaperones

To understand how CdbA depletion results in increased CdbS accumulation and how this over-accumulation may result in disrupted chromosome organization, we searched for protein

interaction partners of CdbA and CdbS using pull-down experiments. An active CdbA-FLAG protein expressed from the native site significantly enriched CdbB and a putative lipoprotein but not CdbS (Fig 4A). This observation agrees with bacterial adenylate cyclase-based two-hybrid (BACTH) analyses demonstrating that CdbA and CdbB interact [11]. Because CdbA is a cytoplasmic protein and the putative lipoprotein is periplasmic, we did not consider this protein further. The active CdbS-FLAG expressed from the native site pulled-down 11 significantly enriched proteins (Fig 4B). These proteins did not include CdbA. These observations support that CdbA and CdbS do not interact directly and that the effect of CdbA-depletion on CdbS accumulation does not involve direct interactions between the two proteins.

Interestingly, the 11 proteins enriched in the CdbS-FLAG pull-down experiment include three chaperones, two co-chaperones and the DnaB helicase (Fig 4B). The remaining five enriched proteins have not previously been analyzed, and none have predicted functions in chromosome organization and protein accumulation. Because CdbA depletion causes defects in chromosome organization and a post-transcriptional upregulation of CdbS accumulation, we focused on DnaB as well as the five chaperones and co-chaperones.

DnaB (Mxan_5084) is the single replicative DNA helicase in *M. xanthus* and is essential for replication initiation [36]. To test whether CdbS and DnaB interact genetically, we first used the temperature-sensitive *dnaB*^A116V mutant, which grows like WT at the permissive temperature but only completes ongoing rounds of DNA replication at the non-permissive temperature at 37°C, and then arrests replication [36]. At 32°C, WT and the *dnaB*^A116V mutant had similar cell lengths and similar chromosome organization patterns (S4A Fig). At 12hrs at 37°, cells of both strains had elongated, but WT cells had mostly well-organized chromosomes, while the *dnaB*^A116V mutant had highly condensed nucleoids centered around midcell and ParB-YFP clusters irregularly localized along the condensed nucleoids (S4A Fig). This chromosome organization differs significantly from the phenotype caused by elevated CdbS levels, with the more decondensed nucleoids organized along the entire cell length. Secondly, we analyzed the *ori/ter* ratio in cells conditionally overexpressing *cdbS-FLAG* from P$_{van}$. Cephalexin-treated WT cells served as a control for elongated cells with normal replication. WT and the *cdbS-FLAG* overexpression strain had no significant differences in their *ori/ter* ratios neither in the absence of vanillate nor in the presence of cephalexin (for 8hrs)/vanillate (for 24hrs) (S4B Fig), supporting that a strain with an elevated CdbS level replicates as WT and that an elevated CdbS level neither inhibits nor stimulates DnaB function. Finally, we used a BACTH analysis with full-length CdbS and DnaB to test for direct interactions. While homo-hexameric DnaB [37], as expected, self-interacted, we did not detect an interaction between CdbS and DnaB (S4C Fig). Thus, how CdbS and DnaB might be connected remains unclear.

The five chaperones and co-chaperones enriched in the CdbS-FLAG pull-down experiments include two DnaK proteins, an Hsp20 protein, a J-domain protein and a GrpE homolog. DnaK proteins, called Hsp70 proteins, are ATP-dependent chaperones that interact directly with their clients. In this way, DnaK proteins contribute to the stability of clients by promoting their *de novo* folding, refolding, or solubilization of aggregates; alternatively, DnaK proteins can also target clients for degradation [38]. DnaK proteins function with two co-chaperones: J-domain proteins bind client proteins and transfer them to the partner DnaK protein, thereby also stimulating DnaK ATPase activity; GrpE proteins are nucleotide exchange factors that stimulate ADP for ATP exchange by their partner DnaK proteins [38]. DnaK proteins also work with small heat shock proteins, e.g. proteins of the Hsp20 family, which are ATP-independent chaperones that bind unfolded or misfolded proteins and transfer them to ATP-dependent chaperones, including DnaK proteins [38].

The DnaK proteins Mxan_3778 and Mxan_6605, which we renamed to CsdK1 and CsdK2 (Cdb<u>S</u> <u>s</u>tabilizing <u>D</u>na<u>K</u>1 and -2, respectively (see below)), are two of 15 *M. xanthus* DnaK

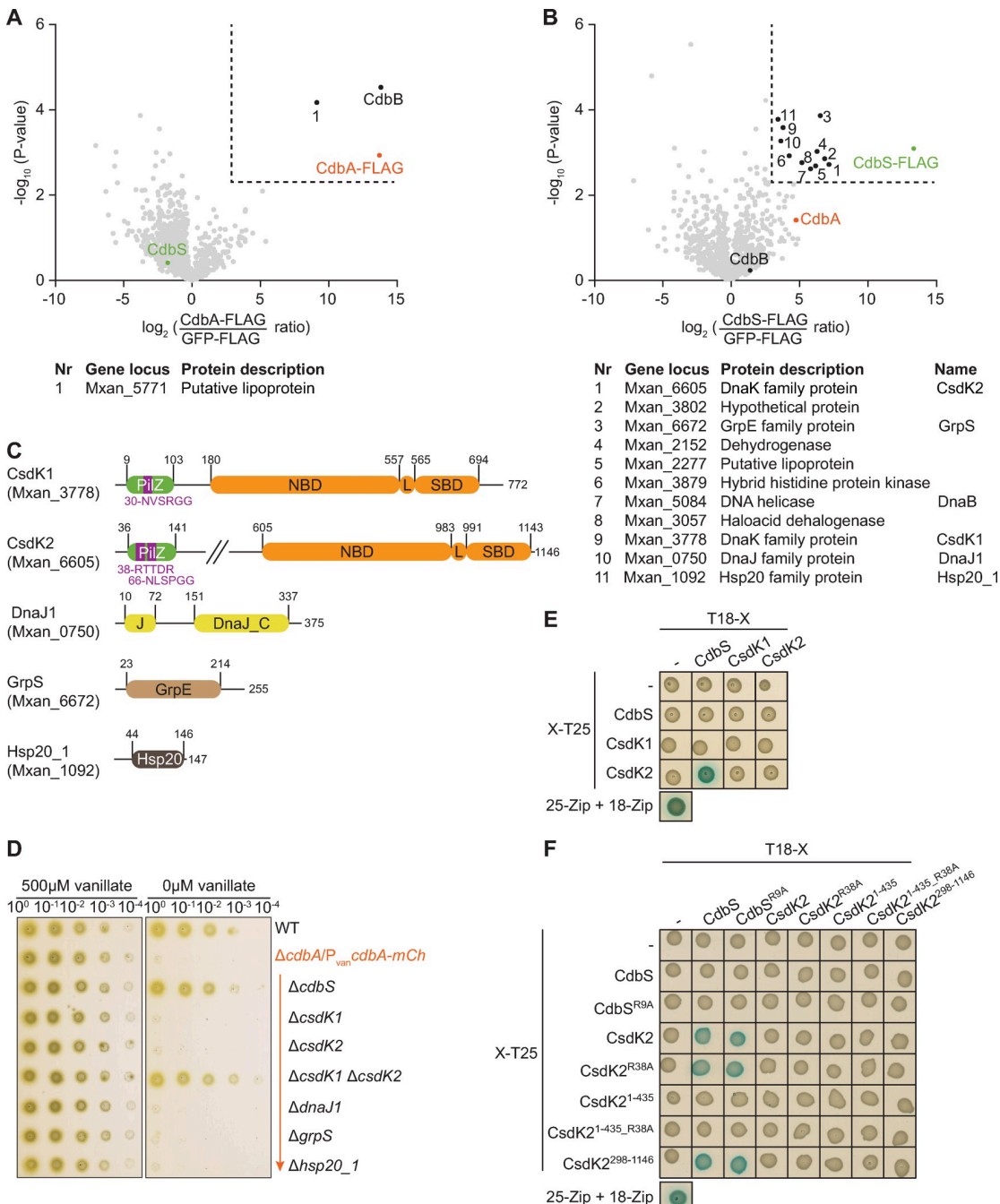

**Fig 4. CdbS interacts with two PilZ-DnaK proteins. A, B.** Volcano plot visualizing potential interaction candidates of CdbA-FLAG (A) and CdbS-FLAG (B). *In vivo* pull-down using CdbA-FLAG or CdbS-FLAG as bait compared to GFP-FLAG (negative control). Samples from four biological replicates were analyzed by label-free mass spectrometry-based quantitative proteomics, and mean iBAQ values and log2-fold enrichment in experimental samples compared to GFP-FLAG samples calculated. X-axis, log2-fold enrichment of proteins with the indicated bait protein versus the control sample expressing GFP-FLAG. Y-axis, -log10 of *P*-value. Significantly enriched proteins in the experimental samples (log2 ratio ≥3; *P*-value ≤0.005 (-log10 ≥2.3) are indicated by the stippled lines. Enriched proteins are indicated or numbered and explained in the tables below. **C.** Domain architecture of chaperones and co-chaperones enriched in the pull-down experiments with CdbS-FLAG. In CsdK1 and CsdK2, residues in purple indicate the c-di-GMP binding motifs (see also Fig 1B), NBD the nucleotide binding domain, L the conserved linker, and SBD the substrate binding domain. **D.** Cells lacking CsdK1 as well as CsdK2 and depleted of CdbA-mCh are viable. Cells were grown on 1% CTT broth, 1.5% agar on a solid surface in the presence and absence of vanillate as indicated. Plates were incubated for 96hrs before imaging. Similar results were observed in three independent experiments. All strains are Δ*cdbA*/ P$_{van}$*cdbA-mCh* and *parB*⁺/*parB-YFP* merodiploid. **E.** BACTH analysis of CdbS and CsdK1 and CsdK2 interactions. The indicated

full-length proteins were fused to the N-terminus of T25 or the C-terminus of T18 as indicated. Blue and white colony colors indicate an interaction and no interaction, respectively. T25-Zip + T18-Zip, positive control; the strains in the row and column labelled "–"contain the indicated plasmid and an empty plasmid and served as controls for self-activation. The same results were observed in two biological replicates. **F.** BACTH analysis of CdbS and CsdK2 interaction. The indicated protein variants were fused to the N-terminus of T25 or the C-terminus of T18 as indicated. Controls as in E. The same results were observed in two biological replicates.

proteins. Both contain an N-terminal PilZ domain and the three core regions of DnaK proteins, i.e. the nucleotide-binding domain, followed by the short conserved linker and the substrate-binding domain (Fig 4C). The CsdK1 PilZ domain lacks the first half of the bipartite c-di-GMP binding motif, while the CsdK2 PilZ domain has the fully conserved c-di-GMP binding motif (Fig 4C), supporting that only this protein might bind c-di-GMP. The nucleotide-binding domains in CsdK1 and CsdK2 show high conservation, while the substrate-binding domains are more diverse (S5 Fig). In the systematic analysis of PilZ domain-containing proteins in *M. xanthus*, CsdK1 and CsdK2 were shown to be dispensable for viability, motility, exopolysaccharide synthesis and development [23].

The DnaJ protein Mxan_0750, which we renamed to DnaJ1, is one of 16 *M. xanthus* J-domain proteins. DnaJ1 contains both the characteristic J-domain and the Gly/Cys-rich DnaJ_C domain, similar to the canonical DnaJ protein of *Escherichia coli* (Figs 4C and S5) [38]. Mxan_6672 is one of two *M. xanthus* GrpE proteins and was identified in previous work, named GrpS and is dispensable for viability and T4P-dependent motility [39]. GrpS contains the conserved GrpE domain (Figs 4C and S7). Mxan_1092, which we renamed Hsp20_1, is one of three *M. xanthus* Hsp20 domain proteins and, similarly to IbpA of *E. coli*, only contains the Hsp20 domain (Figs 4C and S8).

To test genetically whether the five chaperones and co-chaperones are important for the cellular response to CdbA depletion, we generated in-frame deletions of *csdK1*, *csdK2*, *dnaJ1*, *grpS* and *hsp20_1* in the vanillate-dependent CdbA-mCh depletion strain. All five mutations were readily obtained in the presence of vanillate; however, none of the five deletion strains were viable upon CdbA-mCh depletion (Fig 4D). Remarkably, the Δ*csdK1*Δ*csdK2* double mutant was viable upon CdbA-mCh depletion, similar to cells with the Δ*cdbS* mutation upon CdbA-mCh depletion (Fig 4D). Thus, CsdK1 and CsdK2 function redundantly during CdbA depletion and the lack of both proteins, similar to the lack of CdbS, suppresses the lethal CdbA-mCh depletion phenotype.

To test whether CdbS interacts directly with CsdK1 and/or CsdK2, we used a BACTH analysis with full-length CdbS, CsdK1 and CsdK2. We observed an interaction between CdbS and CsdK2 but not between CdbS and CsdK1 (Fig 4E). In a detailed BACTH analysis, CdbS and the non-c-di-GMP binding CdbS$^{R9A}$ variant interacted with the DnaK part of CsdK2 but not with the CsdK2 PilZ domain (Fig 4F). Similarly, the Arg38Ala substitution in the c-di-GMP binding motif in the CsdK2 PilZ domain in the context of either the full-length protein or the PilZ domain did not interfere with the CdbS/CsdK2 interaction (Fig 4F).

These observations support that CdbS, independently of c-di-GMP binding, is a client of CsdK2. Because cells of the Δ*csdK1*Δ*csdK2* double mutant are viable upon CdbA-mCh depletion while cells with the individual deletions are not, we speculate that CdbS is also a client of CsdK1 and that the CdbS/CsdK1 interaction is of too low affinity to be detected in the BACTH analysis. This notion agrees with the CdbS-FLAG pull-down experiments in which CsdK2 was more highly enriched than CsdK1 (Fig 4B).

It remains possible that DnaJ1 and GrpS are co-chaperones of CsdK1 and/or CsdK2 and were enriched in the CdbS-FLAG pull-down experiments because they interact with CsdK1 and/or CsdK2. It is also possible that CbdS interacts directly with GrpS. Similarly, Hsp20_1

might interact directly with CdbS or CsdK1 and/or CsdK2. The observation that the ΔdnaJ1, ΔgrpS and Δhsp20_1 mutations do not suppress the lethal CdbA-mCh depletion phenotype, suggest that if DnaJ1, GrpS and/or Hsp20_1 function together with CdbS, CsdK1 and/or CsdK2, then other J-domain protein(s), the second GrpE homolog and other small heat shock proteins can take over their function.

## CsdK1 and CsdK2 enable the increased CdbS accumulation during CdbA depletion

To investigate how the ΔcsdK1 and ΔcsdK2 mutations jointly suppress the lethal CdbA depletion phenotype, we determined the accumulation of CdbS-FLAG in their presence or absence in a strain in which CdbS-FLAG was synthesized from the native site. CdbS-FLAG accumulated at the same level independently of the two CsdK proteins in the presence of CdbA (Fig 5A). Moreover, all four strains had cell lengths and chromosome organization as WT (S9 Fig). Upon CdbA-mCh depletion without either CsdK1 or CsdK2, the CdbS-FLAG level increased ~4-fold, and cells became filamentous and had a chromosome organization characterized by the irregular distribution of decondensed nucleoids along the cell length and ParB-YFP clusters irregularly distributed over the nucleoid masses and no longer restricted to the subpolar region(s) (Fig 5B and 5C). By contrast, in the absence of both CsdK1 and CsdK2, CdbS-FLAG only accumulated at an ~1.6-fold higher level upon CdbA-mCh depletion, cells only had a slight, although significant, increase in cell length, and most cells had WT-like chromosome organization (Fig 5B and 5C). These findings agree with the observation that cells depleted of CdbA-mCh and lacking CsdK1 and CsdK2 are viable (Fig 4D).

Hence, CsdK1 and CsdK2 are not important for CdbS-FLAG accumulation in the presence of CdbA, but they act redundantly during CdbA-depletion to enable the increased CdbS accumulation. Because this increased CdbS accumulation is post-transcriptionally regulated, and CsdK2 interact with CdbS in the BACTH analysis, we suggest that these two DnaK chaperones stabilize CdbS during CdbA depletion rather than increasing the translation of the cdbS mRNA.

## CdbA-depletion causes increased csdK1 and csdK2 transcription and accumulation

To investigate how CsdK1 and CsdK2 enable the increased CdbS accumulation in response to CdbA depletion, we analyzed the expression of csdK1 and csdK2 and the accumulation of CsdK1 and CsdK2. In RT-qPCR experiments, csdK1 and csdK2 were expressed at ~1.6- and 2.4-fold higher levels, respectively upon CdbA-mCh depletion for 24hrs (Fig 5D). Consistently, the tagged CsdK1-mVenus (from here on CsdK1-mV) and CsdK2-HA variants expressed from their native sites accumulated at ~1.6- and 1.8-fold higher levels upon depletion of CdbA-mCh for 24hrs (Fig 5E and 5F). In published CdbA-FLAG ChIP-seq data [11], the two CdbA binding sites closest to csdK1 are centered at -1976 and +382 relative to the transcription start site at +1, and the two sites closest to csdK2 are centered at -2 and >3500bp away from the transcription start site at +1 (S1 Fig). These observations agree with previous findings that CdbA, even when a binding site is located in a promoter region, only moderately affects transcription [11].

Altogether, these data support a pathway in which CdbA-mCh depletion alleviates the slight repression of csdK1 and csdK2 transcription, resulting in increased CsdK1 and CsdK2 levels that, in turn, stabilize CdbS, thereby enabling its increased accumulation and CdbS toxicity.

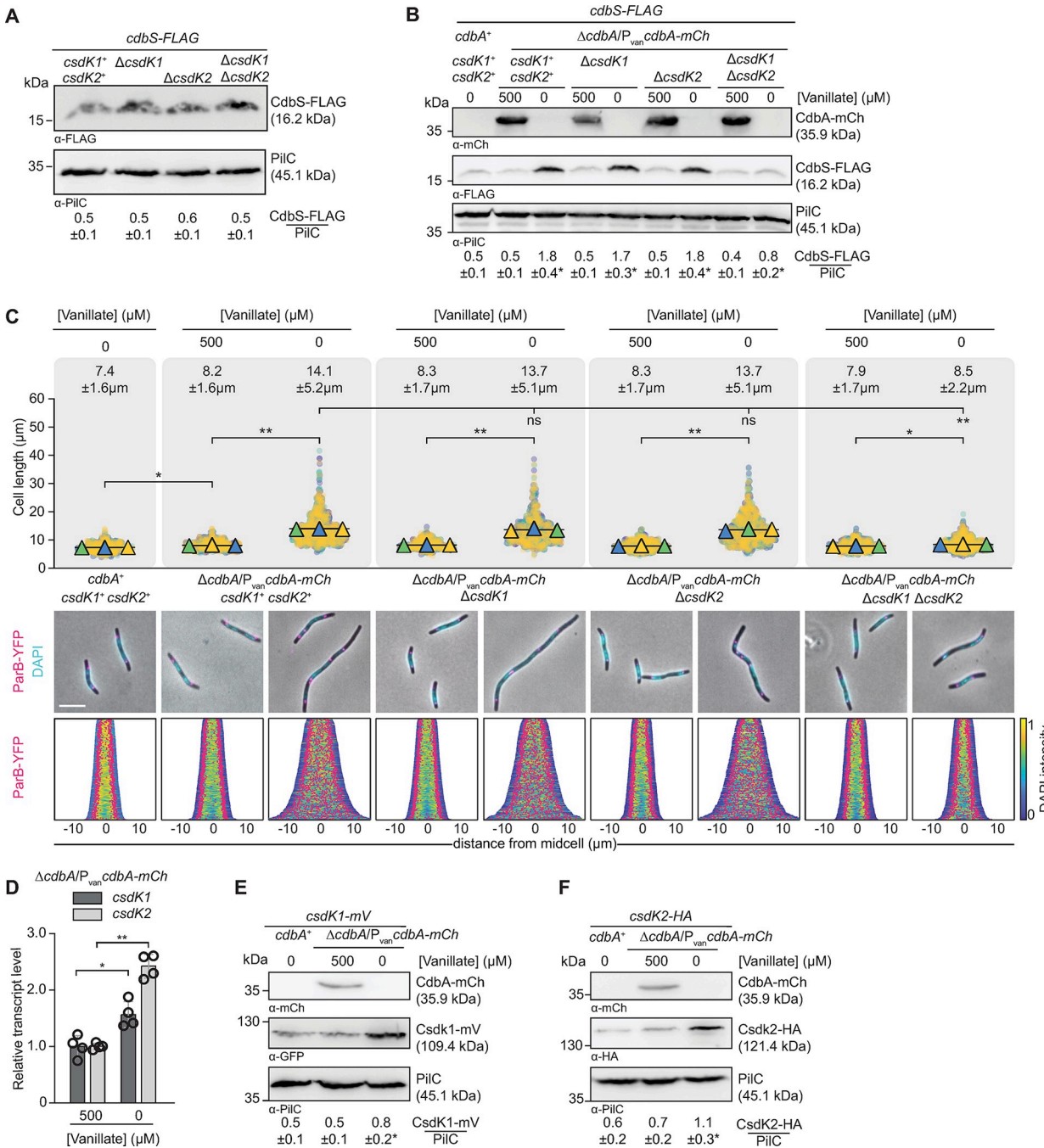

**Fig 5. CsdK1 and CsdK2 jointly stabilize CdbS during CdbA depletion. A.** Immunoblot analysis of CdbS-FLAG accumulation in strains lacking CsdK1, CsdK2 or both. Samples were loaded and analyzed as in Fig 3B based on three biological replicates. In Student's t test in which samples were compared to the *csdK1⁺ csdK2⁺* strain, no significant differences were observed. *cdbS-FLAG* is expressed from the native site. **B.** Immunoblot analysis of CdbS-FLAG accumulation in strains lacking CsdK1, CsdK2 or both and depleted of CdbA-mCh. Cells of the indicated genotypes were grown in the presence of vanillate or in its absence for 24hrs. *cdbS-FLAG* is expressed from the native site. Samples were loaded and analyzed as in Fig 3B based on three biological replicates. *, $P<0.05$ in Student's t test in which samples are compared to the *cdbA⁺* strain. **C.** Cell length and chromosome organization of strains of indicated genotypes. Cells were grown as in B. Cells were analyzed as in Fig 2B. Numbers above indicate cell length as mean ± STDEV from all three experiments. *, $P<0.001$, ** $P<0.0001$, and ns, not significant in 2way ANOVA multiple comparisons test. Total number of cells analyzed: 544–923. Lower diagrams, scale bar, 5μm. N = 400 cells for all strains. **D.** RT-qPCR analysis of *csdK1* and *csdK2* transcript levels in the presence and absence of CdbA-mCh. Cells of the indicated genotypes were grown as in B. Transcript levels are indicated relative to the level in the presence of 500μM vanillate as mean ± STDEV from four biological replicates with three technical replicates each. *, $P<0.01$, **, $P<0.0001$ in two-sided Student's t-test. **E, F.** Immunoblot analysis of CsdK1-mV (E) and CsdK2-HA (F) accumulation in the presence and absence of CdbA-mCh. Cells were grown as in B. Samples were loaded and analyzed as in Fig 3B based on three

biological replicates. *, $P<0.05$ in Student's t test in which samples were compared to the *cdbA*+ strain. In A-D, all strains are *parB*+/*parB-YFP* merodiploid.

## CdbS accumulation increases and accelerates cell death during heat stress

Our genetic analyses establish a pathway for how CdbA depletion results in increased CdbS accumulation with detrimental consequences to cell viability. However, the physiological function of this system is not clear. To address this question, we considered that under the conditions tested, cells lacking CdbS, CdbA and CdbS, or CsdK1 and CsdK2 have no evident differences to WT. We, therefore, speculated that an increased CdbS accumulation reflects the active state of the CdbA/CsdK1/CsdK2/CdbS system. Consequently, we searched for environmental stresses that activate the system using increased CdbS accumulation as a readout for activation. We focused on starvation, temperature and osmotic stress as well as DNA damaging agents because (1) the c-di-GMP level increases 10-fold during development and CdbA, CdbS and possibly CsdK2 bind c-di-GMP, (2) DnaK chaperones are important for protein refolding and stability during environmental stress, and (3) chromosome organization is perturbed by increased CdbS levels.

Using the strain synthesizing CdbS-FLAG from the native site, we found that the CdbS-FLAG level decreased during development (S11A Fig). This observation agrees with neither CdbS, CsdK1 nor CsdK2 being required to complete development [23]. When exposed to different stresses for 18hrs, cells only accumulated CdbS-FLAG at an increased level in response to growth at 37˚C (*M. xanthus* is conventionally grown at 32˚C) but at a reduced level in response to other stresses tested (S11B Fig).

Next, we monitored CdbS-FLAG, CdbA-mCh, CsdK1-mV, and CsdK2-HA accumulation as a function of time at 37˚C using strains that synthesized these proteins from the native sites (Fig 6A). CdbS-FLAG, CsdK1-mV and CsdK2-HA levels increased over time ~3.3-fold, ~1.6-fold and ~2.0-fold compared to 32˚C peaking at 8-18hrs, 8-18hrs and 12-24hrs, respectively. The CdbA-mCh level did not significantly change. Notably, the increased CdbS-FLAG accumulation depended on CsdK1 and CsdK2, and in their absence, CdbS-FLAG only accumulated at a slightly, but significantly, ~1.7-fold higher level at 18hrs at 37˚C (Fig 6A). These protein accumulation profiles are strikingly similar to those observed in response to CdbA depletion.

Consistent with the similarities in the protein accumulation profiles, WT in suspension culture at 37˚C had a lower growth rate than at 32˚C and eventually ceased growth at ~24hrs, followed by a decrease in $OD_{550}$, indicating cell lysis (Fig 6B). Moreover, WT increased in cell length at 37˚C and at 18hrs and later, it had disrupted chromosome organization (Fig 6CD). At 24hrs, many WT cells had lysed, and a significant fraction of cells neither had a DAPI-stained nucleoid nor ParB-YFP clusters indicating extensive chromosome break-down (Fig 6D and 6E). By contrast, the Δ*cdbS* and Δ*csdK1*Δ*csdK2* mutants at 37˚C initially had the same growth rate as at 32˚C, and only ceased growth at ~36hrs followed by a decrease in $OD_{550}$ (Fig 6B). Consistently, they had a less severe filamentation phenotype and no defects in chromosome organization at ~24hrs, and only displayed disrupted chromosome organization and cell lysis lyse at ~36hrs at 37˚C (Fig 6D and 6E). In agreement with these observations, all three strains had a three-log defect in plating efficiency at 37˚C compared to 32˚C (Fig 6B). Finally, a strain expressing the non-c-di-GMP binding CdbS^R9A-FLAG variant phenocopied the WT strain at 37˚C (S12A–S12D Fig). Hence, 37˚C is a lethal growth temperature for *M. xanthus*, and the increased CdbS level mediated by the two CsdK proteins acts during heat stress to accelerate cell death independently of its ability to bind c-di-GMP.

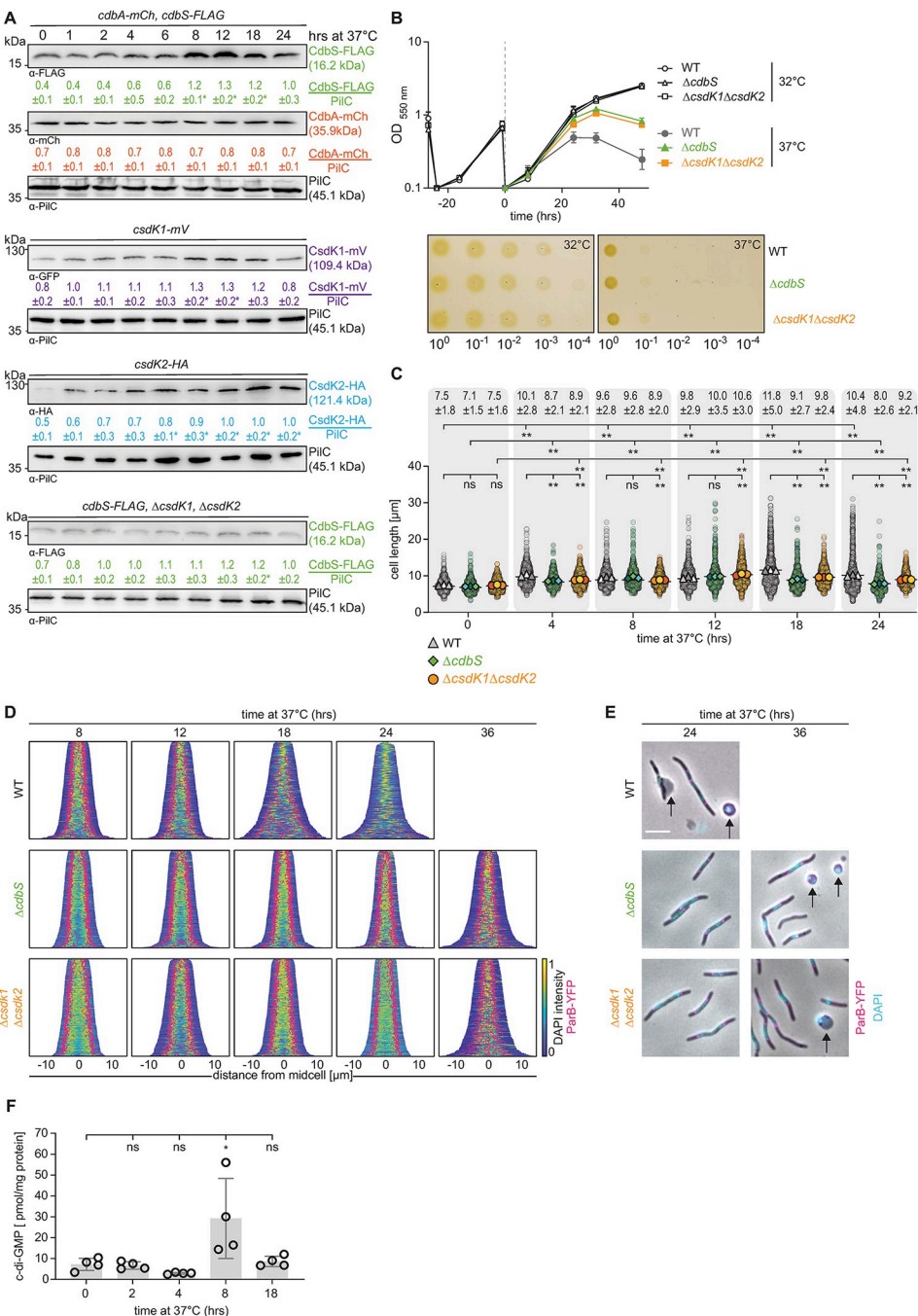

**Fig 6. CdbS accumulates at an increased level at 37˚C and accelerates cell death. A.** Immunoblot analysis of CdbS-FLAG, CdbA-mCh, CsdK1-mV and CsdK2-HA accumulation at 37˚C. Cells of the indicated genotypes were grown at 32˚C and shifted to 37˚C at t = 0hrs. All strains express the relevant protein from the native site. CdbS-FLAG and CdbA-mCh were synthesized in the same strain. Samples were loaded and anylzed as in Fig 3 based on three biological replicates. *, $P<0.05$ in Student's t test in which samples were compared to protein levels at t = 0hrs. **B.** Growth of strains of indicated genotypes. Cells were grown in 1% CTT broth in suspension culture (upper) or on 1% CTT broth, 1.5% agar on a solid surface (lower) at 32˚C and 37˚C as indicated. Growth curves were generated from three independent experiments. Plates were incubated for 96hrs before imaging and similar results were obtained in three biological replicates. **C, D.** Cell length analyses and chromosomes organization in strains of the indicated genotypes during growth at 37˚C. Cells were grown at 37˚C as indicated. In C, cells were analyzed as in Fig 2B. Only rod-shaped cells were included in the measurements. Numbers above indicate cell length as mean ± STDEV calculated from three biological replicates. Total number of cells analyzed: 487–770. *, $P<0.01$, **, $P<0.0001$, ns, not significant in 2way ANOVA multiple comparisons test. In D, only rod-shaped cells were included in the analysis and not cell that

were undergoing lysis or had rounded up (See E) and cells are sorted according to length, DAPI signals are shown according to the intensity scale, and ParB-YFP signals in pink. N = 400 cells for all strains. **E.** Microscopic analysis of cells of the indicated genotypes at 37˚C for the indicated periods. Cells were stained with DAPI (blue signal) and synthesizing ParB-YFP (pink signal). Arrows point to cells undergoing lysis or cells that have lost their rod-shape and have rounded up. Scale bar, 5μm. **F.** c-di-GMP level during growth at 37˚C. Cells were harvested at the indicated time points of incubation at 37˚C, and c-di-GMP levels and protein concentrations determined. Levels are shown as mean ± STDEV calculated from four biological replicates. Individual data points are in black. *, $P<0.05$ in Student's t test. In B-F, all strains are $parB^+/parB\text{-}YFP$ merodiploid.

Interestingly, the CdbA-mCh level did not decrease at 37˚C (Fig 6A). CdbA-mCh co-localized with the nucleoid at both temperatures without evident differences (S13A Fig). Because c-di-GMP and DNA binding by CdbA are mutually exclusive *in vitro* [11], we speculated that an increase in the cellular c-di-GMP level at 37˚C could curb DNA binding by CdbA. Consequently, we determined the cellular level of c-di-GMP at 32˚C and 37˚C. Intriguingly, the c-di-GMP level had increased ~4-fold at 8hrs coinciding with the onset of the increased accumulation of CdbS, CsdK1 and CsdK2 (Fig 6F). At 18hrs, the c-di-GMP level had returned to the level at 32˚C.

Paradoxically, CdbS, CsdK1 and CsdK2 contribute to cell death in cells maintained at 37˚C ≥24hrs. We speculated that these three proteins could contribute a protective function if cells were incubated for shorter periods at 37˚C. To test this idea, WT and ΔcdbS cells were incubated at 37˚C for up to 12hrs and then plated at 32˚C. The WT and ΔcdbS strains progressively lost viability at 37˚C and had the same recovery efficiency at all time points (S13B Fig).

## Discussion

Here we investigated the mechanism underlying the essentiality of the c-di-GMP binding NAP CdbA in *M. xanthus*. We demonstrate that the loss of function of *cdbS*, which encodes a stand-alone PilZ domain protein, completely alleviates the toxicity of the lack of CdbA. Accordingly, CdbA depletion in the presence of CdbS disrupts chromosome organization resulting in inhibition of cell division, cell elongation and, eventually, cell death. By contrast, in the absence of CdbA and CdbS, chromosome organization is not disrupted, and cells are fully viable. Thus, CdbS is the critical factor during CdbA depletion that mediates the disrupted chromosome organization, resulting in cellular filamentation and cell death.

Four key observations indicate a genetic pathway for the link between CdbA and CdbS. First, upon CdbA depletion, CdbS accumulation is upregulated ~4-fold post-transcriptionally. Second, while the two DnaK chaperones CsdK1 and CsdK2 are not important for CdbS accumulation in the presence of CdbA, the ~4-fold increase upon CdbA depletion depends on the redundant activities of CsdK1 and CsdK2. Third, upon CdbA depletion, *csdK1* and *csdK2* transcription increases ~1.6–2.4-fold and, accordingly, the CsdK1 and CsdK2 levels increase ~1.6–1.8-fold. Fourth, a ~4-fold increased CdbS level in the presence of CdbA phenocopies CdbA depletion. These results support a genetic pathway (Fig 7AB) in which CdbA depletion alleviates the slight transcriptional repression of *csdK1* and *csdK2*, resulting in slightly increased CsdK1 and CsdK2 levels. The increased CsdK1 and CsdK2 levels, in turn, enable increased CdbS accumulation. Finally, this ~4-fold increased CdbS accumulation is sufficient for causing disrupted chromosome organization, cellular filamentation and cell death.

Because this genetic pathway is based on gene knock-outs and, thus, the complete loss of function of individual proteins, it could be argued that the effects observed upon CdbA depletion represent an aberrantly activated or overactivated CdbA/CsdK1/CsdK2/CdbS system that may not normally occur in cells [40]. Therefore, we searched for physiological cue(s) that could induce the CdbA/CsdK1/CsdK2/CdbS system. Indeed, the growth of *M. xanthus* at

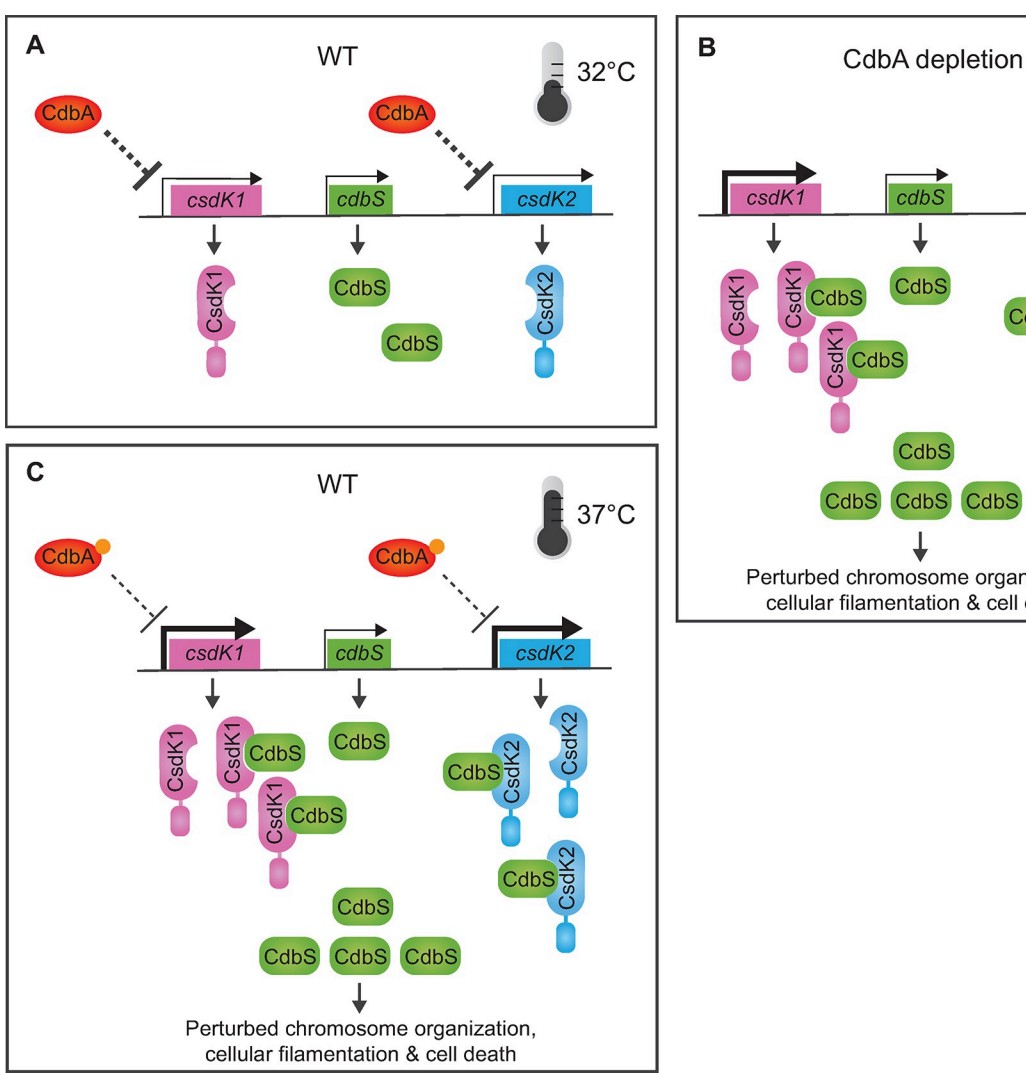

**Fig 7. Schematics of the three states of the CdbA/CsdK1/CsdK2/CdbS system. A.** WT at 32˚C. CdbA slightly represses transcription of *csdK1* and *csdK2;* repression is indicated by dashed line to indicate that it might be indirect via chromosome topology. CdbS, CsdK1 and CsdK2 accumulate, and CsdK1 and CsdK2 are not important for CdbS accumulation. **B.** CdbA depletion at 32˚C. Upon CdbA depletion, repression of *csdK1* and *csdK2* transcription is alleviated, possibly *via* a change in chromosome topology, resulting in increased transcription of the two genes (thick kinked arrows) and increased accumulation of CsdK1 and CsdK2. CsdK1 and CsdK2, in turn, enable increased CdbS accumulation, here shown as a result of the two DnaK proteins stabilizing CdbS. At this elevated level, CdbS perturbs chromosome organization eventually causing cellular filamentation and cell death. **C.** WT at 37˚C. Accumulation of CsdK1 and CsdK2 increases. CsdK1 and CsdK2, in turn, enable increased CdbS accumulation, here also shown as a result of the two DnaK proteins stabilizing CdbS. At this elevated level, CdbS perturbs chromosome organization eventually causing cellular filamentation and cell death. The increased CsdK1 and CsdK2 accumulation is indicated as the result of curbed DNA binding of CdbA caused by an increased level of c-di-GMP (orange dots) that binds to CdbA; as a result, repression of *csdK1* and *csdK2* transcription is alleviated, possibly via a change in chromosome topology, resulting in increased transcription of the two genes (thick kinked arrows).

37˚C (as opposed to the conventional growth temperature at 32˚C) caused increased CsdK1 and CsdK2 accumulation, which, in turn, mediated an increased CdbS accumulation (Fig 7C). At this temperature, the increased CdbS level also contributed to disrupted chromosome organization, cell elongation and cell death. Importantly, at 37˚C, the CdbA accumulation level did not decrease, raising the question of how the CsdK1 and CsdK2 levels increase. Interestingly, we found that the cellular c-di-GMP level increased at 37˚C, coinciding with the onset of the increased accumulation of CdbS, CsdK1 and CsdK2. These observations support the attractive

idea that an increased c-di-GMP level at 37°C could curb DNA binding by CdbA, thereby alleviating the repression of *csdK1* and *csdK2* transcription (Fig 7C). Of note, at 32°C, a *M. xanthus* strain with a c-di-GMP level artificially increased ~7-fold has no defects in chromosome organization, growth and viability [11,20]. Thus, it remains possible that an increased c-di-GMP level at 37°C is sufficient to curb CdbA DNA binding but not at 32°C. Alternatively, the increased accumulation of the two CsdK proteins at 37°C is independent of CdbA or the increased c-di-GMP level functions with a yet-to-be-identified mechanism at 37°C.

Altogether, our data support two pathways for activating the CdbA/CsdK1/CsdK2/CdbS system (Fig 7A–7C). First, in the absence of CdbA, *csdK1* and *csdK2* transcription increases, resulting in increased CsdK1 and CsdK2 levels that stabilize CdbS, thereby enabling its increased accumulation and toxicity (Fig 7B). Second, in response to exposure to 37°C, and possibly involving curbed CdbA DNA binding caused by an increased c-di-GMP level, CsdK1 and CsdK2 levels increase, enabling CdbS stabilization, and thereby its increased accumulation and toxicity (Fig 7C).

Paradoxically, the CdbA/CsdK1/CsdK2/CdbS system contributes to cell death in both pathways, raising the question of this system's physiological function. Generally, a stress response induced by an environmental cue contributes to cellular homeostasis, i.e. after induction by a specific cue, the induced system facilitates cellular adaption to this cue, thereby increasing cellular fitness, and is then switched off. We, therefore, investigated whether the CdbA/CsdK1/CsdK2/CdbS system provides a protective function at 37°C if cells are only incubated for short periods at 37°C, and then returned to 32°C. Under these conditions, WT and Δ*cdbS* cells behaved similarly and progressively lost viability, indicating that the CdbA/CsdK1/CsdK2/CdbS system does not evidently contribute to cellular homeostasis and fitness. Instead, our data suggest that the CdbA/CsdK1/CsdK2/CdbS system may contribute to regulated cell death in the absence of CdbA or at 37°C. Other bacterial systems involved in regulated cell death have in common that they confer an altruistic behavior on the cell undergoing regulated cell death. Such bacterial systems have been described to be part of antiphage mechanisms including dedicated immune systems as well as toxin/antitoxin systems. Upon activation, these systems kill infected cells, thereby preventing spreading of the phage to the remaining population [41–43]. Thus, the infected cell is sacrificed to the benefit of the population [43]. Typically, antiphage immunity systems and toxin/antitoxin systems are encoded by neighboring genes, and in the case of toxin/antitoxin systems consist of two adjacent genes that encode a toxin that causes growth arrest or cell death and a partner antitoxin that counteracts the toxin by direct interaction [41–43]. However, CdbA, CsdK1, CsdK2 and CdbS are not encoded by neighboring genes and with the exception of CdbS and CsdK2, and possibly CsdK1, there is no evidence supporting that they interact. These observations indicate that the CdbA/CsdK1/CsdK2/CdbS system is neither part of an antiphage immunity systems nor a toxin/antitoxin system. Alternatively, a cytoplasmic contractile injection system was recently described in *Streptomyces coelicolor* that contributes to regulated cell death in response to different stresses [44,45]. This system was suggested to cause the release of nutrients by inducing regulated cell death, thereby delaying the starvation-induced formation of spore-filled aerial hyphae [44,45]. However, we previously found that cells lacking CdbS, CsdK1 or CsdK2 form spore-filled fruiting bodies with the same timing and efficiency as WT [23] indicating that the function of the CdbA/CsdK1/CsdK2/CdbS system is not to regulate the timing of development. Thus, the precise function of the CdbA/CsdK1/CsdK2/CdbS system remains an open question and will be important to address in the future.

We confirmed that *M. xanthus* cells elongate when grown at 37°C [46–48]; however, in these previous experiments, cells were only followed for 12-18hrs at this temperature. A surprising finding from our investigations is that under our conditions, 37°C is a lethal growth

temperature for *M. xanthus* independently of CdbS. However, the CdbA/CsdK1/CsdK2/CdbS system accelerates cell death at 37˚C.

CdbA depletion causes disrupted chromosome organization that, in turn, inhibits cell division causing cellular filamentation and cell death. Here we show that these effects of CdbA depletion depend on CdbS. In other words, it is not the lack of CdbA that causes the massive defects in chromosome organization; the increased CdbS level causes these defects. Because CdbA binds >550 sites on the *M. xanthus* chromosome with moderate sequence specificity, is highly abundant and has only minor effects on transcription (here; [11]), CdbA fulfils the criteria for being a NAP [15–17]. Therefore, we suggest that the effect of CdbA depletion, and possibly also in response to exposure to 37˚C, on transcription of *csdK1* and *csdK2* is likely indirect *via* changes in chromosome topology (Fig 7B and 7C). Equally, the observation that DNA binding and c-di-GMP binding are mutually exclusive [11] supports that DNA binding is modulated by c-di-GMP [11]. However, the essential function of CdbA is to maintain the CdbS level appropriately low.

How, then, does a high CdbS level disrupt chromosome organization? Lack of CdbS in the presence or absence of CdbA affects neither the chromosome nor viability. Thus, CdbS is not essential for these two processes. In CdbS-FLAG pull-down experiments, the replicative DnaB helicase was enriched. However, our results do not provide support for a direct interaction between CdbS and DnaB. Interestingly, the PlzA protein of *Borrelia burgdorferi*, which consists of two PilZ domains connected by a short linker that binds c-di-GMP [49], was reported to bind DNA and RNA in a c-di-GMP-dependent manner [50]. As opposed to PlzA [50], CdbS consists of a single PilZ domain, and its function is independent of c-di-GMP binding. Nevertheless, it remains possible that CdbS could be a DNA/RNA-binding protein, thereby contributing to chromosome organization. The two main contributors to chromosome organization and segregation in *M. xanthus*, i.e. the ParABS system and SMC, are essential at 32˚C [29,33]. Thus, it is also possible that increased CdbS levels interfere with the functioning of one or both of these systems. It will be important to address whether CdbS binds DNA/RNA in the future. Also, the pull-down experiments with CdbS-FLAG were conducted under conditions where CdbS was not overaccumulating (and, thus, not toxic). Therefore, seeking potential interaction partners under conditions where CdbS is overaccumulating and toxic to cells will also be important.

The two DnaK proteins CsdK1 and CsdK2 function redundantly to mediate the increased CdbS accumulation in response to CdbA depletion and exposure to 37˚C. These observations, taken together with the results of the CdbS-FLAG pull-down experiments and BACTH analyses in which CdbA interacts with the core DnaK regions of CsdK2 but not with the CsdK2 PilZ domain, support that CdbS is a client of CsdK1 and CsdK2. CsdK1 and CsdK2 are only important for the increased accumulation in response to CdbA depletion and heat stress at 37˚C but not for CdbS accumulation in the presence of CdbA at 32˚C. Because the increased CdbS accumulation in response to CdbA depletion is regulated post-transcriptionally, we suggest that the two CsdK proteins stabilize CdbS upon CdbA depletion and heat stress at 37˚C, likely by stimulating correct folding of CdbS (Fig 7B and 7C).

Intriguingly, among the four CdbA/CsdK1/CsdK2/CdbS system proteins, CdbA and CdbS are verified c-di-GMP binding proteins, and the CsdK2 PilZ domain likely binds c-di-GMP. C-di-GMP and DNA binding by CdbA are mutually exclusive *in vitro*; by contrast, c-di-GMP binding by CdbS is not important for toxicity under the conditions tested; and, finally, it is not known whether c-di-GMP binding by the CsdK2 PilZ domain is important for function. As described, we speculate that the increased c-di-GMP level at 37˚C could modulate DNA binding by CdbA, thereby contributing to the increased accumulation of the two CsdK proteins at 37˚C. In *M. xanthus*, c-di-GMP has previously been shown to regulate T4P-dependent motility

and extracellular matrix composition [20,22,23] as well as fruiting body formation and sporulation [21,24]. *M. xanthus* is predicted to encode 11 enzymatically active diguanylate cyclases and six enzymatically active phosphodiesterases [20,21]. Among these, the verified diguanylate cyclases DmxA and DmxB are specifically important during growth and development, respectively and the verified phosphodiesterase PmxA is specifically important for development [20,21,24]. By contrast, lack of any the remaining 14 predicted enzymes interferes with neither motility nor development [20,21] and their function remains unknown. Interestingly, growth temperature has been implicated in regulating the c-di-GMP level in other bacteria. A low temperature causes increased c-di-GMP accumulation in *Vibrio cholerae* and *Pseudomonas putida* by unknown mechanisms [51]. In *P. aeruginosa*, an increased temperature causes an increased c-di-GMP level via direct activation of the thermosensitive diguanylate cyclase TdcA [52]. In the future, it will be interesting to investigate the mechanism underlying the increased c-di-GMP level in response to exposure of *M. xanthus* cells to 37°C, to test whether any of the diguanylate cyclases and/or phosphodiesterases of unknown function are involved, and to determine how the increased c-di-GMP level might contribute to regulated cell death.

C-di-GMP is ubiquitous in bacteria and typically associated with regulation of lifestyle transitions [1,2]. However, it is increasingly recognized that c-di-GMP is also involved in regulating very different types of processes including modification of ribosomal proteins in *P. fluorescens* [53] and metabolic flux in *E. coli* [54]. Based on the data presented here, we propose that c-di-GMP might also be involved in regulated cell death.

## Materials and methods

### Strains and cell growth

All *M. xanthus* strains are derivatives of the WT DK1622 [55] and are listed in Table 2. Plasmids and primers used are listed in Table 3 and S1 Table, respectively. All numerical data are included in S2 Table. In-frame deletions were generated as described [56]. Plasmids for ectopic expression of genes were integrated in a single copy either by site-specific recombination into the Mx8 *attB* site or by homologous recombination at the *mxan_18/19* locus. All plasmids were verified by DNA sequencing and all strains were verified by PCR. *M. xanthus* was grown at 32°C in 1% CTT broth (1% Bacto Casitone (Gibco), 10mM Tris-HCl pH 8.0, 1mM KPO$_4$ pH 7.6, 8mM MgSO$_4$) [57] or on 1.5% agar supplemented with 1% CTT broth, and kanamycin (50µg mL$^{-1}$) or oxytetracycline (10µg mL$^{-1}$) if relevant. Mitomycin C and nalidixic acid were added to final concentrations of 20µg mL$^{-1}$. Plasmids were propagated in *E. coli* NEB Turbo ((F' *proA$^+$B$^+$ lacI$^q$ ΔlacZM15/fhuA2 Δ(lac-proAB) glnV galK16 galE15 R(zgb-210::Tn10)* Tet$^S$ *endA1 thi-1 Δ(hsdS-mcrB)5*)) (New England Biolabs) at 37°C in lysogeny broth (LB) [58] supplemented with kanamycin (50µg mL$^{-1}$), tetracycline (15µg mL$^{-1}$) or carbenicillin (100µg mL$^{-1}$).

### Development

Cells were developed under submerged conditions as described [59]. Briefly, exponentially growing *M. xanthus* in CTT broth were harvested at 5,000 *g* for 5min and resuspended in MC7 buffer (10 mM 3-(*N*-morpholino)propanesulfonic acid (MOPS) pH 6.8, 1 mM CaCl$_2$) to $7\times10^9$ cells mL$^{-1}$. 50µL was added to 350µL of MC7 buffer in 24-well polystyrene plate (Falcon) and incubated at 32°C.

### Whole genome sequencing

Chromosomal DNA of the eight suppressor mutants and the original strain (SA5691) was isolated using the MasterPure DNA Purification Kit (Epicentre Biotechnologies) according to the

**Table 2. *M. xanthus* strains used in this study.**

| Strain | Genotype | Reference |
|---|---|---|
| DK1622 | WT | [55] |
| SA5691 | Δ*cdbA*; *mxan18-19*::P$_{van}$*cdbA-mCh*; *parB$^+$/attB*::P$_{nat}$*parB-YFP* | [11] |
| SA5693 | *parB$^+$/attB*::P$_{nat}$*parB-YFP* | [11] |
| SA8813 | *cdbA*::*cdbA-FLAG* | [11] |
| SA11494 | *attB*::P$_{pilA}$*gfp-FLAG* | This study |
| SA10209 | Δ*cdbA*; *mxan18-19*::P$_{van}$*cdbA-mCh*; *cdbS*::*cdbS-FLAG*; *parB$^+$/attB*::P$_{nat}$*parB-YFP* | This study |
| SA10217 | *cdbS*::*cdbS-FLAG*; *parB$^+$/attB*::P$_{nat}$*parB-YFP* | This study |
| SA10220 | Δ*cdbS*; *parB$^+$/attB*::P$_{nat}$*parB-YFP* | This study |
| SA10225 | Δ*cdbS*; *mxan18-19*::P$_{van}$*cdbS-FLAG*; *parB$^+$/attB*::P$_{nat}$*parB-YFP* | This study |
| SA10226 | Δ*cdbS*; *mxan18-19*::P$_{van}$*cdbS$^{R9A}$-FLAG*; *parB$^+$/attB*::P$_{nat}$*parB-YFP* | This study |
| SA10249 | *cdbS*::*cdbS-FLAG*; Δ*csdK1*; *parB$^+$/attB*::P$_{nat}$*parB-YFP* | This study |
| SA10251 | *cdbS*::*cdbS-FLAG*; Δ*csdK2*; *parB$^+$/attB*::P$_{nat}$*parB-YFP* | This study |
| SA10260 | *cdbA*::*cdbA-mCh*; *parB$^+$/attB*::P$_{nat}$*parB-YFP* | This study |
| SA10262 | Δ*cdbA*; *mxan18-19*::P$_{van}$*cdbA-mCh*; *cdbS*::*cdbS-FLAG*; Δ*csdK1*; *parB$^+$/attB*::P$_{nat}$*parB-YFP* | This study |
| SA10264 | Δ*cdbA*; *mxan18-19*::P$_{van}$*cdbA-mCh*; *cdbS*::*cdbS-FLAG*; Δ*csdK2*; *parB$^+$/attB*::P$_{nat}$*parB-YFP* | This study |
| SA10267 | *cdbS*::*cdbS-FLAG*; Δ*csdK1*; Δ*csdK2*; *parB$^+$/attB*::P$_{nat}$*parB-YFP* | This study |
| SA10270 | Δ*cdbA*; *mxan18-19*::P$_{van}$*cdbA-mCh*; *cdbS*::*cdbS-FLAG*; Δ*grpS*; *parB$^+$/attB*::P$_{nat}$*parB-YFP* | This study |
| SA10273 | Δ*cdbA*; *mxan18-19*::P$_{van}$*cdbA-mCh*; *cdbS*::*cdbS-FLAG*; Δ*hsp20_1*; *parB$^+$/attB*::P$_{nat}$*parB-YFP* | This study |
| SA10274 | *cdbA*::*cdbA-mCh*; *cdbS*::*cdbS-FLAG*; *parB$^+$/attB*::P$_{nat}$*parB-YFP* | This study |
| SA10275 | *cdbA*::*cdbA-mCh*; Δ*cdbS*; *mxan18-19*::P$_{van}$*cdbS-FLAG*; *parB$^+$/attB*::P$_{nat}$*parB-YFP* | This study |
| SA10277 | Δ*cdbA*; *mxan18-19*::P$_{van}$*cdbA-mCh*; *cdbS*::*cdbS-FLAG*; Δ*csdK1*; Δ*csdK2*; *parB$^+$/attB*::P$_{nat}$*parB-YFP* | This study |
| SA10288 | *csdK1*::*csdK1-mV* | This study |
| SA10289 | *csdK2*::*csdK2-HA* | This study |
| SA12204 | Δ*cdbA*; *mxan18-19*::P$_{van}$*cdbA-mCh*; *csdK1*::*csdK1-mV* | This study |
| SA12205 | Δ*cdbA*; *mxan18-19*::P$_{van}$*cdbA-mCh*; *csdK2*::*csdK2-HA* | This study |
| SA12206 | Δ*cdbA*; *mxan18-19*::P$_{van}$*cdbA-mCh*; Δ*cdbS*; *parB$^+$/attB*::P$_{nat}$*parB-YFP* | This study |
| SA12209 | Δ*cdbA*; *mxan18-19*::P$_{van}$*cdbA-mCh*; *cdbS*::*cdbS-FLAG*; Δ*dnaJ1*; *parB$^+$/attB*::P$_{nat}$*parB-YFP* | This study |
| SA12222 | Δ*cdbA*; *mxan18-19*::P$_{van}$ *cdbA-mCh*; Δ*cdbS*; *attB*::P$_{nat}$ *cdbS-FLAG*::P$_{nat}$*parB-yfp* | This study |
| SA12238 | *dnaB$^{A116V}$*; *parB$^+$/attB*::P$_{nat}$*parB-YFP* | [36] and this study |

manufacturer's recommendation. The FS libraries were prepared and sequenced (paired-end, 2×250 bp) on an Illumina HiSeq2500 instrument at the Max Planck-Genome-Centre Cologne. CLC workbench 12.0 (Qiagen, Hilden, Germany) was used for computational processing of sequencing data.

## Immunoblots

Immunoblotting was performed as described in [58]. For sample preparation, *M. xanthus* cells were harvested from suspension cultures or from cells developed under submerged conditions and resuspended in sodium dodecyl sulfate (SDS) lysis buffer. The same amount of protein was loaded per sample (30μg). After electrophoresis, proteins were transferred to a nitrocellulose membrane (Cytiva) with 0.2μm pore size using transfer buffer (300mM Tris-HCl, 300mM

**Table 3. Plasmids used in this study.**

| Plasmid | Description | Reference |
|---|---|---|
| pBJ114 | *galK*, Kan$^R$ | [76] |
| pET28a(+) | Expression vector, Kan$^R$ | Merck (Darmstadt) |
| pMR3691 | *mxan18-19* site integration, *vanR*-P$_{van}$, Tet$^R$ | [30] |
| pSW105 | Mx8 *attB* integration, P$_{pilA}$, Kan$^R$ | [77] |
| pSWU19 | Mx8 *attB* integration, Kan$^R$ | [78] |
| pSWU30 | Mx8 *attB* integration, Tet$^R$ | [79] |
| pUT18 | BACTH plasmid, *cyaAT18* N-terminal fusion, Amp$^R$ | Euromedex (Souffelweyersheim, France) |
| pUT18C | BACTH plasmid, *cyaAT18* C-terminal fusion, Amp$^R$ | Euromedex (Souffelweyersheim, France) |
| pKNT25 | BACTH plasmid, *cyaAT25* N-terminal fusion, Kan$^R$ | Euromedex (Souffelweyersheim, France) |
| pKT25 | BACTH plasmid, *cyaAT25* N-terminal fusion, Kan$^R$ | Euromedex (Souffelweyersheim, France) |
| pDJS94 | pBJ114; Δ*csdK2*, Kan$^R$ | [23] |
| pDJS151 | pSWU19; P$_{nat}$*parB-YFP*, Kan$^R$ | [11] |
| pMAT219 | pSW105; *gfp-FLAG*, Kan$^R$ | [80] |
| pSK42 | pBJ114; Δ*csdK1*, Kan$^R$ | [23] |
| pMS007 | pET28a(+); His$_6$-*cdbS*, Kan$^R$ | This study |
| pMS008 | pET28a(+); His$_6$-*cdbS*$^{R9A}$, Kan$^R$ | This study |
| pMS018 | pBJ114; *cdbS-FLAG*, Kan$^R$ | This study |
| pMS024 | pMR3691; P$_{van}$*cdbS-FLAG*, Tet$^R$ | This study |
| pMS026 | pMR3691; P$_{van}$*cdbS*$^{R9A}$-*FLAG*, Tet$^R$ | This study |
| pMS054 | pBJ114; *cdbA-mCh*, Kan$^R$ | This study |
| pMS055 | pBJ114; Δ*dnaJ1*, Kan$^R$ | This study |
| pMS056 | pBJ114; Δ*grpS*, Kan$^R$ | This study |
| pMS057 | pBJ114; Δ*hsp20_1*, Kan$^R$ | This study |
| pMS088 | pBJ114; *csdK1-mV*, Kan$^R$ | This study |
| pMS089 | pBJ114; *csdK2-HA*, Kan$^R$ | This study |
| pMS102 | pSWU19; P$_{nat}$*parB-YFP*; P$_{nat}$*cdbS-FLAG*; Kan$^R$ | This study |
| pKNT25-CdbS | pKNT25; *cdbS*, Kan$^R$ | This study |
| pKT25-CdbS | pKT25; *cdbS*, Kan$^R$ | This study |
| pUT18-CdbS | pUT18; *cdbS*, Amp$^R$ | This study |
| pUT18C-CdbS | pUT18C; *cdbS*, Amp$^R$ | This study |
| pKT25-CsdK1 | pKT25; *csdK1*, Kan$^R$ | This study |
| pUT18C-CsdK1 | pUT18C; *csdK1*, Amp$^R$ | This study |
| pKT25-CsdK2 | pKT25; *csdK2*, Kan$^R$ | This study |
| pUT18C-CsdK2 | pUT18C; *csdK2*, Amp$^R$ | This study |
| pKNT25-DnaB | pKNT25; *dnaB*, Kan$^R$ | This study |
| pKT25-DnaB | pKT25; *dnaB*, Kan$^R$ | This study |
| pUT18-DnaB | pUT18; *dnaB*, Amp$^R$ | This study |
| pUT18C-DnaB | pUT18C; *dnaB*, Amp$^R$ | This study |
| pKT25-CdbS$^{R9A}$ | pKT25; *cdbS*$^{R9A}$, Kan$^R$ | This study |
| pUT18C-CdbS$^{R9A}$ | pUT18C; *cdbS*$^{R9A}$, Amp$^R$ | This study |
| pKT25-CsdK2$^{R38A}$ | pKT25; *csdK2*$^{R38A}$, Kan$^R$ | This study |
| pUT18C-CsdK2$^{R38A}$ | pUT18C; *csdK2*$^{R38A}$, Amp$^R$ | This study |
| pKT25-CsdK2$^{1-435}$ | pKT25; *csdK2*$^{1-435}$, Kan$^R$ | This study |
| pUT18C-CsdK2$^{1-435}$ | pUT18C; *csdK2*$^{1-435}$, Amp$^R$ | This study |
| pKT25-CsdK2$^{1-435\_R38A}$ | pKT25; *csdK2*$^{1-435\_R38A}$, Kan$^R$ | This study |
| pUT18C-CsdK2$^{1-435\_R38A}$ | pUT18C; *csdK2*$^{1-435\_R38A}$, Amp$^R$ | This study |
| pKT25-CsdK2$^{298-1146}$ | pKT25; *csdK2*$^{298-1146}$, Kan$^R$ | This study |
| pUT18C-CsdK2$^{298-1146}$ | pUT18C; *csdK2*$^{298-1146}$, Amp$^R$ | This study |

glycine, 0.05% (w/v) SDS, pH 9.0). As primary antibodies, rabbit polyclonal α-FLAG (1:2,000, Rockland), α-mCherry (1:2500, BioVision), α-HA (1:2000, Sigma) and α-PilC (1:2000) [60]) were used with horseradish-peroxidase-conjugated α-rabbit immunoglobulin G (1:15000, Sigma) as secondary antibody. α-GFP (1:2000, Sigma) primary antibodies were used together with horseradish peroxidase-conjugated α-mouse immunoglobulin G (1:5000, GE Healthcare) as secondary antibody. α-polyHistidine antibodies conjugated with peroxidase (1:2000, Sigma) were used to detect $His_6$-tagged proteins. Immunoblots were developed using Forte Western HRP Substrate (Millipore) on a LAS-4000 imager (Fujifilm). The signal intensities of the individual bands were quantified using Fiji [61]. Each band intensity of the protein of interest was normalized relative to the intensity of the respective PilC loading control, mean ± STDEV were calculated from three independent biological replicates.

## RT-qPCR

Total RNA from *M. xanthus* cells grown on 1.5% agar supplemented with 1% CTT broth was extracted using the Monarch Total RNA Miniprep Kit (New England Biolabs). Briefly, $10^9$ cells were scraped off the agar-plates and resuspended in 200μL lysis-buffer (100mM Tris-HCl pH 7.6, 1mg mL$^{-1}$ lysozyme), and incubated at 25°C, 5min. RNA purification was performed according to manufacturer's protocol. DNA was removed using Turbo DNase (Thermo Fisher Scientific) and DNase was removed using the Monarch RNA Cleanup Kit (50 μg; New England Biolabs). LunaScript RT SuperMix Kit (New England Biolabs) was used to generate cDNA using 1μg RNA. qPCR reactions were performed on an Applied Biosystems 7500 Real-Time PCR system using the Luna Universal qPCR MasterMix (New England Biolabs) with the primers listed in S1 Table. *rpsS*/*mxan_3303*, which encodes the small ribosomal subunit protein S19, was used as an internal control [62]. Data analysis was performed using the comparative $C_T$ method [63].

## Microscopy

Fluorescence microscopy was performed as described [11]. Briefly, exponentially growing cells were stained with 1mg mL$^{-1}$ DAPI for 10min at 32°C, transferred to a pad containing 1.5% agarose (Cambrex) with TPM buffer (10 mM Tris-HCl pH 7.6, 1 mM $KPO_4$ pH 7.6, 8 mM $MgSO4$) supplemented with 0.2% CTT broth on a microscope slide, and covered with a coverslip. A Leica DMi8 inverted microscope was used for imaging, and phase contrast and fluorescence snapshots were acquired using a Hamamatsu ORCA-flash V2 Digital CMOS camera. For image processing, Metamorph v 7.5 (Molecular Devices) was used. Using a custom made Matlab R2020a (MathWorks) script, cells and fluorescent signals were detected automatically using Oufti48 [64].

## Determination of *ori*/*ter* ratio

To determine the *ori*/*ter* ratio, chromosomal DNA was isolated as described. The *ori* region was amplified using the primer pairs ori 1/dnaA fwd with ori 1/dnaA rev and ori 2/7483 fwd with ori 2/7483 rev, while the *ter* region was amplified using the primer pair 3778_ter 1 qPCR fwd with 3778_ter 1 qPCR rev and the Luna Universal qPCR Master Mix (New England Biolabs). Quantification of the *ori*/*ter* ratio was performed using the the $2^{-\Delta CT}$ method as described [65].

## Protein purification

To purify $His_6$-CdbS and $His_6$-CdbS$^{R9A}$, *E.coli* Arctic Express (DE3) RP (Agilent Technologies) was transformed with pMS007 and pMS008, respectively. Cultures were grown in 2L LB

with gentamycin and kanamycin to an $OD_{600}$ of 0.5 to 0.7 at 30°C. Protein overproduction was induced by adding isopropyl-β-D-1-thiogalactopyranoside (IPTG) to a final concentration of 0.5 mM, and then the culture was grown at 11°C for 24hrs. Cells were harvested by centrifugation at 3,800 $g$ for 30min at 4°C. The pellet was resuspended in 25mL Lysis Buffer (50mM Tris-HCl pH 7, 150mM NaCl, 10% (v/v) glycerol, 1mM ß-mercaptoethanol, 10mM imidazole, cOmplete protease inhibitor EDTA-free (Roche Diagnostics GmbH)) and sonicated for 30min with a UP200St sonifier (60% pulse, 50% amplitude, 30sec on/off time; Hielscher) on ice. The solution was centrifuged at 48,000 $g$ at 4°C for 45min at 4°C. The soluble fraction was loaded onto a 5ml HiTrap Chelating HP column (GE Healthcare) that had been pre-loaded with $NiSO_4$ according to the manufacturer's recommendation and pre-equilibrated with wash buffer (50mM Tris-HCl pH 7, 150mM NaCl, 10% (v/v) glycerol, 1mM 1mM ß-mercaptoethanol, 20mM imidazole). The column was washed with 10 column volumes wash buffer. Proteins were eluted with elution buffer (50mM Tris-HCl pH 7, 150mM NaCl, 10% (v/v) glycerol, 1mM ß-mercaptoethanol, 500mM imidazole) with a linear imidazole gradient from 50 to 500mM. Fractions containing the protein of interest were combined and mixed with anion exchange buffer A (50mM Tris-HCl pH 7, 10% (v/v) glycerol, 1mM ß-mercaptoethanol) in a 1:3 ratio and then loaded onto a 5 ml HiTrap SP HP (GE Heathcare) column. Bound protein was eluted along a 5 column volume gradient of anion exchange buffer B (50mM Tris-HCl pH 7, 10% (v/v) glycerol, 1mM ß-mercaptoethanol, 2M NaCl). Fractions containing the protein of interest were pooled and concentrated using an Ultra-4 3K centrifugal filter unit (Merck) according to the manufacturer's recommendation, while the buffer was exchanged to dialysis buffer (50mM Tris-HCl pH 7, 150mM NaCl, 10% (v/v) glycerol). Purified protein aliquots were shock-frozen in liquid nitrogen and stored at -80°C.

### Bio-layer interferometry

Bio-layer interferometry was performed using the BLItz system (forteBio) as described [12]. 500nM biotinylated c-di-GMP (Biolog) in dialysis buffer supplemented with 0.02% (wt/vol) Tween-20 was loaded onto a Streptavidin SA biosensor (forteBio) for 120sec followed by 30sec of washing with dialysis buffer. To calculate binding kinetics, varying concentration of the protein of interest in dialysis buffer supplemented with 0.02% (wt/vol) Tween-20 was applied to the biosensor for 120sec followed by 120sec of dissociation with dialysis buffer.

### Determination of c-di-GMP level

The c-di-GMP level was determined as described [24]. Briefly, cultures were harvested at 2,500 $g$ for 20min at 4°C. After lysing cells in extraction buffer (high-pressure liquid chromatography [HPLC]-grade acetonitrile-methanol-water [2/2/1, vol/vol/vol]), the supernatants were evaporated to dryness in a vacuum centrifuge. Subsequently, the pellets were dissolved in HPLC-grade water and analyzed by liquid chromatography-tandem mass spectrometry (LC-MS/MS). The c-di-GMP level was measured at the Research Service Centre Metabolomics at the Hannover Medical School, Germany.

### *In vivo* pull-down and label-free mass spectrometry-based quantitative proteomics

Exponentially growing cultures were harvested by centrifugation at 2,500 $g$ at 20°C for 10min. The pellet was resuspended in 10ml HNN buffer (50mM HEPES pH 7.2, 150mM NaCl, 5mM EDTA, cOmplete protease inhibitor (Roche Diagnostics GmbH), 0.5% (v/v) NP40) and sonicated for 1min with a UP200St sonifier (60% pulse, 50% amplitude; Hielscher) on ice. To each sample, 10µl anti-FLAG M2 magnetic beads (Merck) were added. Next, the samples were

placed in an overhead rotor for 90min at 4°C. The supernatant was removed and the beads were washed with HNN buffer followed by four times washing with 100mM ammoniumbicarbonate to remove all detergent and protease inhibitors. Further sample processing was carried out as described [66]. Briefly, enriched proteins were eluted by adding 1μg trypsin (Promega) and incubation for 30min at 30°C, and further incubated overnight in the presence of 5mM Tris(2-carboxyethyl)phosphin (TCEP). Following, acetylation using 10mM iodoacetamide for 30min at 25°C in the dark, the peptides were desalted using C18 solid phase extraction. Liquid chromatography-mass spectrometry (LC-MS) analysis of the peptide samples were carried out on a Q-Exactive Plus instrument connected to an Ultimate 3000 RSLCnano and a nanospray flex ion source (all Thermo Scientific). Peptide separation was performed on a C18 reverse phase HPLC column (75μm × 42cm; 2.4μm, Dr. Maisch). The peptides were loaded onto a PepMap 100 precolumn (Thermo Scientific) and eluted by a linear acetonitrile (ACN) gradient from 6–35% solvent B over 30min (solvent A: 0.15% formic acid; solvent B: 99.85% ACN in 0.15% formic acid) with 300nL min$^{-1}$ flow rate. The spray voltage was set to 2.5kV, and the temperature of the heated capillary was set to 300°C. Survey full-scan MS spectra (m/z = 375–1500) were acquired in the Orbitrap with a resolution of 70,000 (at m/z 200) after accumulation a maximum of $3 \times 10^6$ ions in the Orbitrap. Up to 10 most intense ions were subjected to fragmentation using high collision dissociation (HCD) at 27% normalized collision energy. Fragment spectra were acquired at 17,500 resolution. The ion accumulation time was set to 50ms for both MS survey and MS/MS scans. The charged state screening modus was enabled to exclude unassigned and singly charged ions. The dynamic exclusion duration was set to 30sec.

Label-free quantification of the samples was performed using MaxQuant (Version 1.6.10.43) [67]. For Andromeda database searches implemented in the MaxQuant environment, a *M. xanthus* Uniprot protein database (downloaded in 10/2016) was used. The search criteria were set as follows: full tryptic specificity was required (cleavage after lysine or arginine residues); two missed cleavages allowed; carbamidomethylation (C) was set as fixed modification; oxidation (M) and deamidation (N, Q) as variable modification. MaxQuant was operated in default settings without the "Match-between-run" option. For protein quantification, iBAQ values (intensity-based absolute quantification) were calculated within MaxQuant [68]. Calculated iBAQ values were normalized to iBAQ-protein sum of all detected proteins. Student's t-test was performed within Perseus [69] with the following parameters (FDR: 0.01, s0: 0.5).

The proteomics data have been deposited to the ProteomeXchange consortium via the PRIDE partner repository [70] with the dataset identifier PXD041344.

## Bioinformatics

The structural model of CdbS was generated using AlphaFold via ColabFold [71,72] using the Alphafold2_mmseqs2 notebook with default settings, except recycles were set to six; figure of protein model was generated using PyMOL (Schrödinger LLC). Predicted Local Distance Difference Test (pLDDT) and predicted Alignment Error (pAE) graphs of the five models generated by Alphafold2_mmseqs2 notebook were made using a custom made Matlab R2020a (The MathWorks) script. Ranking of the models was performed based on combined pLDDT and pAE values, with the best-ranked model used for further analysis and presentation. Protein domains were predicted with HMMER [73]. Alignments and phylogenies were constructed using MEGA-X [74]. Similarity and identity between proteins were analyzed with EMBOSS Needle [75].

## Plasmid construction

Genomic *M. xanthus* DNA or plasmids were used as templates to amplify genes and genomic regions of interest.

pMS007 and pMS008 are derivatives of pET28a(+): *cdbS* was amplified from genomic DNA using primers 4328 fw_NdeI and oMS004. To generate the *cdbS*[R9A] allele, *cdbS* was amplified from genomic DNA using the primers oMS002 and oMS004. The resulting fragment was used as a template for another PCR and amplified using the primers 4328 fw_NdeI and oMS004. Both inserts were inserted into pET28a(+)via the NdeI/HindII sites.

pMS018 is a derivative of pBJ114: To amplify the upstream fragment, the primers oMS024 and oMS013 were used using genomic DNA. To generate the downstream fragment, the primer pairs oMS025 and oMS087 were used. The final fragment was obtained via overlap PCR and inserted into pBJ114 via the HindIII/EcoRI sites.

pMS024 and pMS026 are derivatives of pMR3691: To amplify FLAG-tagged *cdbS*, the gene was amplified using the primers 4328 fw_NdeI and oMS041 from SA10217. The fragment was inserted into pMR3691 via the NdeI/EcoRI sites. To amplify the *cdbS*[R9A] allele, the primer pairs oMS002 and oMS041 were used with pMS024 as a template. The resulting fragment was used for another PCR using 4328 fw_NdeI and oMS041 as primers. This final fragment was inserted into pMR3691 via the NdeI/EcoRI sites.

pMS054 is a derivative of pBJ114: To amplify the upstream fragment, primers oMS082 and oMS083 were used with genomic DNA. To amplify the mCherry fragment, the primer pairs oMS084 and oMS085 were used using SA5691 as a template. These two fragments were ligated via the BamHI site. To generate the downstream fragment, the primer pairs oMS086 and oMS087 were used with genomic DNA as a template. The final fragment was obtained via overlap PCR and inserted into pBJ114 via the HindIII/EcoRI sites.

pMS055: To amplify the upstream fragment, the primers oMS092 and oMS093 were used using genomic DNA as a template. To generate the downstream fragment, the primer pairs oMS094 and oMS095 using genomic DNA as a template. The final fragment was obtained via overlap PCR and was inserted into pBJ114 via the HindIII/EcoRI sites.

pMS056: To amplify the upstream fragment, the primers oMS100 and oMS101 were used using genomic DNA as a template. To generate the downstream fragment, the primer pairs oMS102 and oMS103 using genomic DNA as a template. The final fragment was obtained via overlap PCR and was inserted into pBJ114 via the HindIII/EcoRI sites.

pMS057 is a derivative of pBJ114: To amplify the upstream fragment, the primers oMS116 and oMS117 were used using genomic DNA as a template. To generate the downstream fragment, the primer pairs oMS118 and oMS119 using genomic DNA as a template. The final fragment was obtained via overlap PCR and was inserted into pBJ114 via the HindIII/EcoRI sites.

pMS088: To amplify the upstream fragment, the primers oMS177 and oMS231 were used using genomic DNA as a template. To amplify the mVenus fragment, the primers oMS194 and oMS232 were used with pFM60 (gift of Franziska Müller) as template. To generate the downstream fragment, the primer pairs oMS228 and oMS180 using genomic DNA as a template. The upstream fragment and the mVenus fragment were fused via overlap PCR, before the resulting fragment was fused to the downstream fragment via a second overlap PCR. This final fragment was inserted into pBJ114 via the HindIII/EcoRI sites.

pMS089: To amplify the upstream fragment, the primers oMS182 and oMS183 were used using genomic DNA as a template. To generate the downstream fragment, the primer pairs oMS184 and oMS185 using genomic DNA as a template. The final fragment was obtained via overlap PCR and was inserted into pBJ114 via the HindIII/EcoRI sites.

pMS102: The $P_{nat}parB$-*YFP* fragment was amplified from pDJS151 using the primers oMS068 and oMS211. The $P_{nat}cdbS$-*FLAG* fragment was amplified from genomic DNA using the primer pairs oMS212 and oMS004. The two fragments were ligated via the NdeI site, and the resulting fragment was inserted into pBJ114 via the XbaI/EcoRI sites.

BACTH plasmids were all constructed using the same strategy. For most constructs, the same fragment of each gene was amplified from genomic DNA and inserted into the respective plasmids via the XbaI/KpnI. For CdbS constructs, the primers oMS131 and oMS132 were used, while for the CdbS$^{R9A}$ variant, the template was pMS026. For CsdK1 constructs, the primers oMS133 and oMS134 were used. For DnaB constructs, the primers oMS154 and oMS155 were used. For CsdK2 constructs in pUT18C, the primers oMS135 and oMS136 were used. For the pKT25-CsdK2 construct, the primer pair oMS135 and oMS141 was used. All CsdK2 fragments were inserted into the respective vector via the XbaI/EcoRI sites.

For the CsdK2$^{R38A}$ variants, the primer pairs oMS135 and oMS204 and additionally oMS205 and oMS206 were used to amplify two fragments that were then fused using overlap PCR. The resulting fragment was cloned into the BACTH plasmids containing CsdK2 using the XbaI/XmaI sites.

For the CsdK2$^{1-435}$ variant, the primer oMS135 was used together with oMS208 for the pKT25 construct or oMS206 for pUT18C constructs. Similarly, the same primer pairs were used for the CsdK2$^{1-435\_R38A}$ variants using pUT18C-CsdK2R38A as a template. For the CsdK2$^{298-1146}$ variant, the primer oMS207 was used together with oMS141 for the pKT25 construct or oMS136 for pUT18C constructs.

## Supporting information

**S1 Fig. Analysis of the *cdbS* locus and the CdbS protein. A.** *cdbS* locus. Upper diagram, transcription direction is indicated by the orientation of arrows, kinked arrows indicate transcription start sites as mapped in [24]. Coordinates indicate bp relative to the transcription start site of *cdbS*. Red triangle indicates the CdbA peak summit from a ChIP-seq analysis in which an active CdbA-FLAG protein was used as bait [11]. The lower diagram show data from RNAseq as base-by-base alignment coverage for total RNA isolated from cells growing in 1% CTT broth [24]. Positive and negative values indicate reads mapped to the forward and reverse strand, respectively. Reads assigned to a gene are colored according to the gene color code in the upper diagram; intergenic regions are in gray. Numbers in genes show mxan_ locus-tags. **B.** The *cdbS* locus is conserved in myxobacteria. Transcription direction is indicated by the orientation of arrows with the color used in A. CdbS homologs were identified using reciprocal BLASTP analysis. Numbers indicated % similarity/identity between CdbS of *M. xanthus* and homologs. % similarity/identity were calculated using EMBOSS Needle software (pairwise sequence alignment). **C.** Alignment of CdbS proteins. Proteins were aligned with default parameters in MEGA7. Amino acid substitution/stop codon caused by *cdbS* suppressor mutations are indicated in red.
(TIF)

**S2 Fig. AlphaFold model of CdbS. A.** pLDDT (Predicted Local Distance Difference Test) and pAE (predicted Alignment Error) plots for five models of CdbS (A) as predicted by AlphaFold. Model rank 1 was used for further analysis and is shown below colored based on pLDDT. **B.** SDS-PAGE analysis of purified His$_6$-CdbS proteins used *in vitro*. 1μg of the indicated purified proteins were separated by SDS-PAGE and gels stained with InstantBlue and corresponding immunoblot analysis with α-His6 antibodies.
(TIF)

**S3 Fig. c-di-GMP binding by CdbS is not important for its activity *in vivo*. A.** Immunoblot analysis of CdbS*-FLAG accumulation. Cells of the indicated genotypes were grown in the presence or absence of the indicated concentrations of vanillate. Cells grown in the presence of vanillate were analyzed 24hrs after addition of vanillate. The *cdbS-FLAG* strain expresses this

allele from the native site. The same amount of total protein was loaded per lane. PilC is used as a loading control. Numbers below show the mean ± STDEV of CdbS-FLAG normalized by the PilC level calculated from three independent experiments. *, $P<0.05$ in Student's t test in which samples were compared to CdbS-FLAG expressed from the native site. All strains are $parB^+$/$parB$-$YFP$ merodiploid. **B.** Cell length and chromosome organization of strains of indicated genotypes. Cells were grown in 1% CTT broth in the presence and absence of vanillate as indicated. Cells grown in the presence of vanillate were analyzed 24hrs after addition of vanillate. Cell length measurements are included from three independent experiments indicated in different colored triangles and the mean based on all three experiments. Numbers above indicate cell length as mean ± STDEV from all three experiments. ns, not significant in 2way ANOVA multiple comparisons test. Total number of cells analyzed: 469–643. Lower panels, fluorescence microscopy images of cells stained with DAPI and synthesizing ParB-YFP. In the demographs, cells are sorted according to length, DAPI signals are shown according to the intensity scale, and ParB-YFP signals in pink. Scale bar, 5μm. N = 400 cells for all strains. All strains are $parB^+$/$parB$-$YFP$ merodiploid.
(TIF)

**S4 Fig. High CdbS levels do not affect DNA replication. A.** Cell length and chromosome organization of strains of indicated genotypes. Cells were grown at 37°C for 12hrs before the analysis. Cell length measurements are from three independent experiments indicated in different colored triangles and the mean is based on all three experiments. Numbers above indicate cell length as mean ± STDEV from all three experiments. ** $P<0.0001$, ns, not significant in 2way ANOVA multiple comparisons test. Total number of cells analyzed: 603–769. Lower diagrams, fluorescence microscopy images of cells stained with DAPI and synthesizing ParB-YFP. In the demographs, cells are sorted according to length, DAPI signals are shown according to the intensity scale, and ParB-YFP signals in pink. Scale bar, 5μm. N = 400 cells for all strains. All strains are $parB^+$/$parB$-$YFP$ merodiploid. **B.** qPCR analysis of $ori/ter$ ratio in indicated strains. Left diagram, the positions on the *M. xanthus* chromosome of the primers used for determination of the $ori/ter$ ratio. Right diagram, ratios are shown relative to the level in untreated WT as mean ± STDEV from four biological replicates with three technical replicates each. In pairwise comparisons, no significant differences were observed in 2way ANOVA multiple comparisons test. Both strains are $parB^+$/$parB$-$YFP$ merodiploid. **C.** BACTH analysis of CdbS and DnaB interaction. The indicated proteins were fused to the N-terminus and C-terminus of T25 or the N- and C-terminus of T18 as indicated. Blue and white colony colours indicate an interaction and no interaction, respectively. T25-Zip + T18-Zip, positive control; the strains in the row and column labelled "–"contain the indicated plasmid and an empty plasmid and served as controls for self-activation. The same results were observed in two biological replicates.
(TIF)

**S5 Fig. Sequence analysis of CsdK1 and CsdK2.** Alignment of the DnaK characteristic domains of CsdK1 and CsdK2 with those of DnaK proteins of *E. coli* and *C. crescentus*. The nucleotide-binding domain (orange), the linker (pink) and the substrate-binding domain (blue) are indicated.
(TIF)

**S6 Fig. Sequence analysis of DnaJ1.** Alignment of full-length DnaJ1 with DnaJ of *E. coli* and *C. crescentus*. The J domain (yellow) and the J_central domain (turquoise) are indicated.
(TIF)

**S7 Fig. Sequence analysis of GrpS.** Alignment of the GrpE domain of GrpS with those of GrpE of *E. coli* and *C. crescentus*. The GrpE domain (brown) is indicated.
(TIF)

**S8 Fig. Sequence analysis of Hsp20_1.** Alignment of the Hsp20 domain of Hsp20_1 with those of IbpA of *E. coli* and the domain of CC_3592 of *C. crescentus*. The Hsp20 domain (cyan) is indicated.
(TIF)

**S9 Fig. Lack of CsdK1 and CsdK2 do not affect cell length and chromosome organization.** Cell length and chromosome organization of strains of indicated genotypes. Cell length measurements are included from three independent experiments indicated in different colored triangles and the mean calculated based on all three experiments. Numbers above indicate cell length as mean ± STDEV from all three experiments. ns, not significant in 2way ANOVA multiple comparisons test. Total number of cells analyzed: 540–719. Lower diagrams, fluorescence microscopy images of cells stained with DAPI and expressing ParB-YFP. In the demographs, cells are sorted according to length, DAPI signals are shown according to the intensity scale, and ParB-YFP signals in pink. Scale bar, 5μm. N = 400 cells for all strains. All strains are *parB*⁺/*parB-YFP* merodiploid.
(TIF)

**S10 Fig.** Organization of the csdK1 (A) and csdK2 (B) loci. A, B. Upper diagrams, transcription direction is indicated by the orientation of arrows, kinked arrows indicate transcription start sites as mapped in [24]. Coordinates indicate bp relative to the transcription start site of *csdK1* and *csdK2*, respectively. The lower diagrams show data from RNAseq as base-by-base alignment coverage for total RNA isolated from cells growing in 1% CTT broth [24]. Positive and negative values indicate reads mapped to the forward and reverse strand, respectively. Reads assigned to a gene are colored according to the gene color code in the upper diagrams. Red triangles indicate the CdbA peak summits from a ChIP-seq analysis in which an active CdbA-FLAG protein was used as bait [11].
(TIF)

**S11 Fig. Immunoblot analysis of CdbS-FLAG accumulation under different stress conditions. A.** CdbS-FLAG accumulation decreases during development. Cells were developed under submerged conditions and harvested at the indicated time points. The same amount of protein was loaded per lane. PilC was used as a loading control. Similar results were obtained in two independent experiments. **B.** CdbS-FLAG accumulates at an increased level at 37˚C. Cells were exposed to the indicated stresses for 18hrs and then harvested. The same amount of protein was loaded per lane. PilC was used as a loading control. Similar results were obtained in two biological replicates. CdbS-FLAG was synthesized from the native *cdbS* locus. All samples were loaded on the same gel; gaps indicate lanes removed for presentation purposes.
(TIF)

**S12 Fig. CdbS accelerates cell death at 37˚C independently of c-di-GMP binding. A.** Growth of strains of indicated genotypes. Cells were grown in 1% CTT broth in suspension culture at the indicated temperatures. Growth curves were generated from three biological replicates. All strains are *parB*⁺/*parB-YFP* merodiploid. **B, C.** Cell length analyses (B) and chromosomes organization (C) in strains of the indicated genotypes during growth at 37˚C. Cells were grown at 37˚C for indicated periods. In B, cell length measurements are included from three independent experiments indicated in different colored triangles and the mean is based on all three experiments. Numbers above indicate cell length as mean ± STDEV calculated

from all three experiments. *, $P < 0.05$, **, $P < 0.01$ and ns, not significant in 2way ANOVA multiple comparisons test. Only rod-shaped cells were included in the measurements. Total number of cells analyzed: 421–938. In C, only rod-shaped cells were included in the analysis and not cell that were undergoing lysis or had rounded up (See E) and cells are sorted according to length, DAPI signals are shown according to the intensity scale, and ParB-YFP signals in pink. N = 400 cells for all strains. **D.** Microscopic analysis of cells of the indicated genotypes at 37˚C for the indicated period. Cells were stained with DAPI (blue signal) and synthesizing ParB-YFP (pink signal). Arrows point to cells that have lost their rod-shape and rounded up. Scale bar, 5 μm. A-D, both strains are *parB*⁺/*parB-YFP* merodiploid.
(TIF)

**S13 Fig. Analysis of CdbA and CdbS at 37˚C. A.** CdbA-mCh colocalizes with the nucleoid at 37˚C. Cells of the indicated genotype were incubated at 37˚C for the indicated periods. Phase contrast and fluorescence microscopy images of cells of the indicated genotype stained with DAPI and synthesizing CdbA-mCh. Scale bar, 5μm. CdbA-mCh was synthesized from the native *cdbA* locus. The strain is *parB*⁺/*parB-YFP* merodiploid. **B.** CdbS does not provide a protective function at 37˚C. Cells of the indicated genotypes were incubated at 37˚C for the indicated periods and then plated at 32˚C. Data represent the mean ± STDEV of three biological replicates normalized to the number of colony forming units at t = 0hrs at 37˚C (100%). In pairwise comparisons, no significant differences were observed in Student's t-test.
(EPS)

**S1 Table. Primers used in this study.**
(DOCX)

**S2 Table. Source data file with all numerical data.**
(XLSX)

# Acknowledgments

We thank Penelope Higgs, Lee Kroos and Daniel Wall for helpful discussions and Anke Treuner-Lange, Sabrina Huneke and Franziska Müller for strains and plasmids.

# Author Contributions

**Conceptualization:** Michael Seidel, Dorota Skotnicka, Lotte Søgaard-Andersen.

**Data curation:** Michael Seidel, Dorota Skotnicka, Timo Glatter.

**Formal analysis:** Michael Seidel, Dorota Skotnicka, Timo Glatter.

**Funding acquisition:** Lotte Søgaard-Andersen.

**Investigation:** Michael Seidel, Dorota Skotnicka, Timo Glatter.

**Project administration:** Lotte Søgaard-Andersen.

**Supervision:** Lotte Søgaard-Andersen.

**Validation:** Michael Seidel, Dorota Skotnicka, Timo Glatter.

**Visualization:** Michael Seidel, Lotte Søgaard-Andersen.

**Writing – original draft:** Michael Seidel, Timo Glatter, Lotte Søgaard-Andersen.

**Writing – review & editing:** Michael Seidel, Dorota Skotnicka, Timo Glatter, Lotte Søgaard-Andersen.

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
