## [Decision Letter · Decision Letter 0]

8 May 2023

Dear Dr Søgaard-Andersen,

Thank you very much for submitting your Research Article entitled 'During heat stress in Myxococcus xanthus, the CdbS PilZ domain protein, along with two PilZ-DnaK chaperones, perturbs chromosome organization and accelerates cell death' to PLOS Genetics.

The manuscript was fully evaluated at the editorial level and by independent peer reviewers. All three reviewers stated that this is an interesting and well-executed study but raised some questions that we ask you address in a revised manuscript. Specifically, all reviewers have asked for clarification and/or additional discussion of the possible function of CdbS and the role of CdG in this process. Reviewer 2 (point 4) has suggested an experiment that may clarify the proposed model, which you could consider.

We ask you to modify the manuscript according to the review recommendations. Your revisions should address the specific points made by each reviewer.

Yours sincerely,

Sean Crosson

Section Editor

PLOS Genetics

Reviewer's Responses to Questions

**Comments to the Authors:**

Reviewer #1: The manuscript by Seidel et al., describes a follow up study of the work of Skotnicka et al, published in Nature Communications in 2020. Previously, the authors identified CdbA as an essential c-di-GMP binding protein and address here the question why CdbA is essential in Myxococcus xanthus. Using an accurate suppressor screen, they identify CdbS - a protein of previously undefined function - which mediates cell toxicity upon CdbA deletion. They confirm using a set of mutants and overexpression strains that CdbS accumulates upon CdbA depletion and causes defects in chromosome organization and consequently cell death. Moreover, they provide detailed mechanistic understanding of how and why CdbS accumulates when CdbA levels decrease. Using a pull-down assay they identify two chaperons, CsdK1 and CsdK2, the expression of which is increased in the cdbA depletion strain and which interact with CdbS and stabilize the protein. Moreover, the authors show that heat stress (growth at 37°C) induces the CdbS/CsdK1+2 system and contributes to cell death.

I really enjoyed reading this manuscript. The experiments are very well performed, important controls are presented and the manuscript is very well written. The work provides many mechanistic insights and covers a very interesting field on NAPs and PilZ-domain containing proteins. Many bacteria e.g. Vibrio cholerae have multiple PilZ domains, but our knowledge about the physiological role of these domains is still very limited.

The only aspect which I personally found a bit “disappointing” is that there are no hints about what could be the function of CdbS and how this protein might affects chromosome organization. Obviously, the authors are aware about this limitation and discuss this point and I understand that this question can only be addressed in a follow-up study.

There are only few minor comments that I would like to make:

1. The authors often use the term “disrupted chromosome organization” and I do not fully understand what exactly is meant here. In line 259, they say “nucleoid as well as ParB-YFP cluster localization were highly disorganized” and refer to Fig. 3DE. I find it very difficult to see what the authors mean in the images provided. Maybe some zoom-in images can be provided to make this point more clearly?

2. The cascade described here covers many different players so I was wondering whether the authors can provide something like a summary image that covers the relevant components and signals. This would certainly be helpful for better understanding of the reader.

3. Line 156: sentence that starts with “By immunoblot…”- a verb is missing.

Reviewer #2: This manuscript by Seidel et. al. explores the molecular mechanism of essentiality of a cyclic di-GMP (c-di-GMP) binding nuclear-associated protein CdbA. To understand why this protein is essential, the authors performed suppressor analysis and found that lethality is due to a new c-di-GMP binding protein, CdbS. The authors find that CdbS accumulates when CdbA is depleted due to the chaperones CsdK1/CsdK2. The authors uncover many new molecular pathways, although strangely they do not seem connected to c-di-GMP, even though all of these proteins bind to c-di-GMP (with the exception of CsdK1). They also importantly show that this pathway is activated during growth at 37 deg C, although they do not give any plausible explanation for how this could be adaptive (See point 5). The data are impressive and solid and support the author’s conclusions. They answer many questions about this system, but significant questions remain such as how CdbS causes cell death and the role of c-di-GMP. However, I appreciate they cannot figure out all aspects of this complex system in one manuscript. But I do think testing their proposed model for increased c-di-GMP triggering this system at 37 deg C would be within the scope of this manuscript (point 4). In sum, this is an interesting, well done paper that should be impactful to the microbiology community.

1. Fig. S4B would be more convincing with a positive control that alters the ori/ter ratio. A thymine auxotroph mutant that becomes starved for thymine would suffice.

2. Line 398-400-I think it reasonable to propose that CsdK1 and CsdK2 stabilize CdbS, but I don’t understand the argument that because accumulation of CdbS occurs post-transcriptionally, this rules out that these proteins do not impact translation.

3. One experiment that is missing that would significantly support their model is to overexpress CdbS in the CsdK1/CsdK2 cdA-mCh strain. If these protein function solely through enhancing CdbS levels, then overexpression of this protein should restore sensitivity to vanillate depletion.

4. Lines 463-470-The authors propose a model whereby high temperatures induce the activity of CdbS not by a decrease in CdbA but by an increase in c-di-GMP concentrations. But the authors do not test this model, which seems straightforward and would be a nice cap to the manuscript. One way they could test this model is by increasing temperature in the c-di-GMP binding mutant of CdbA, which should not have the death phenotype at higher temperatures.

5. Lines 540-547-I appreciate the authors do not understand why this systems promotes death at 37 deg Celsius, but they do not offer any reasonable explanation or speculation. Death would only be favored if it contributes to fitness of related neighbors. Why would this be the case at 37 deg C? Perhaps this phenotype is media specific and only observed in the lab in rich growth media whereas more physiologically relevant environments for Myxo would reveal the fitness benefit of activation of this pathway.

Reviewer #3: The manuscript by Seidel et al. identified a PilZ-like protein, CdbS, which when stabilized by two chaperones, CsdK1 and CsdK2, accelerate cell death under heat stress. This work revealed why CdbA, a nucleoid-associated protein, is essential for M. xanthus. Overall, this manuscript contains substantial data which are logically organized and presented. I have only a few minor comments.

1. The title. Exactly how the over-accumulated CdbS triggers cell death is unknown. I do agree with the authors that the phenotype of CdbS over-accumulation is related to the mis-organization of chromosome. However, there is no evidence that CdbS, or CsdK1 and CsdK2, directly perturb chromosome. Thus, the title is misleading.

2. It is hard to believe that although CdbS and CsdK2 bind c-di-GMP, their functions are not affected by c-di-GMP at all. The role of c-di-GMP needs to be (at least) better discussed. What is the estimated copy number of CdbS? Rather than being an unlikely "toxin/antitoxin" partner for CdbA, could CdbS function as a sink for c-di-GMP, which promotes CdbA to bind DNA and thus slows down the over-accumulation of CdbS?

3. Some small grammar issues. Line 587, the increased accumulation of CdbS. Line 593: c-di-GMP binding by the ...

**Have all data underlying the figures and results presented in the manuscript been provided?**

Reviewer #1: Yes

Reviewer #2: Yes

Reviewer #3: Yes

PLOS authors have the option to publish the peer review history of their article (what does this mean?). If published, this will include your full peer review and any attached files.

Reviewer #1: No

Reviewer #2: No

Reviewer #3: **Yes: **Beiyan Nan

---

## [Decision Letter · Decision Letter 1]

7 Jun 2023

Dear Dr Søgaard-Andersen,

We are pleased to inform you that your manuscript entitled "During heat stress in Myxococcus xanthus, the CdbS PilZ domain protein, in concert with two PilZ-DnaK chaperones, perturbs chromosome organization and accelerates cell death" has been editorially accepted for publication in PLOS Genetics. Congratulations!

Yours sincerely,

Sean Crosson

Section Editor

PLOS Genetics

Sean Crosson

Section Editor

PLOS Genetics

Comments from the reviewers (if applicable):

Reviewer's Responses to Questions

**Comments to the Authors:**

Reviewer #1: Congratulations to the authors - very nice study!

Reviewer #2: The manuscript by Seidel et. al. uncovers significant new biology in Myxococcus xanthus that further increases our understanding of the essentiality of the CbdA nuclear associated protein. I won’t restate the summary from my first review. Overall, the manuscript is strong. But it is a bit disappointing that the authors did not seem to seriously consider my suggested experiments. I appreciate that the experiments suggested as is may not be feasible, but there were other potential ways to address the concerns. For example, for a positive control for Fig. S4B they could have examined the ts DnaB-A116V mutant that they use in their manuscript. This should show altered ori/ter ratios at the non-permissive temperatures. The impetus for suggesting such a control was just to validate the experiment, making the data stronger. To test the model proposed that the elevated c-di-GMP at 37 degrees C inhibits CbdA to drive CbdS mediated killing, the authors could have measured cell killing while overexpressing a phosphodiesterase. Nevertheless, I did not consider any of these experiments essential for publication and have no further suggestions or comments.

Reviewer #3: The authors have addressed all my questions.

**Have all data underlying the figures and results presented in the manuscript been provided?**

Reviewer #1: Yes

Reviewer #2: Yes

Reviewer #3: None

PLOS authors have the option to publish the peer review history of their article (what does this mean?). If published, this will include your full peer review and any attached files.

Reviewer #1: No

Reviewer #2: No

Reviewer #3: **Yes: **Beiyan Nan

**Data Deposition**

http://datadryad.org/submit?journalID=pgenetics&manu=PGENETICS-D-23-00436R1

**Press Queries**

---

## [Editor Report · Acceptance letter]

13 Jun 2023

PGENETICS-D-23-00436R1 

During heat stress in Myxococcus xanthus, the CdbS PilZ domain protein, in concert with two PilZ-DnaK chaperones, perturbs chromosome organization and accelerates cell death 

Dear Dr Søgaard-Andersen, 

We are pleased to inform you that your manuscript entitled "During heat stress in Myxococcus xanthus, the CdbS PilZ domain protein, in concert with two PilZ-DnaK chaperones, perturbs chromosome organization and accelerates cell death" has been formally accepted for publication in PLOS Genetics! Your manuscript is now with our production department and you will be notified of the publication date in due course.

With kind regards,

Zsofia Freund

PLOS Genetics

On behalf of:
